# Amyloidogenic proteins in the SARS-CoV and SARS-CoV-2 proteomes

Taniya Bhardwaj[1], Kundlik Gadhave [1], Shivani K. Kapuganti[1], Prateek Kumar[1], Zacharias Faidon Brotzakis[2], Kumar Udit Saumya[1], Namyashree Nayak[1], Ankur Kumar[1], Richa Joshi[1], Bodhidipra Mukherjee[1], Aparna Bhardwaj[1], Krishan Gopal Thakur [3], Neha Garg [4], Michele Vendruscolo [2] & Rajanish Giri [1] ✉

The phenomenon of protein aggregation is associated with a wide range of human diseases. Our knowledge of the aggregation behaviour of viral proteins, however, is still rather limited. Here, we investigated this behaviour in the SARS-CoV and SARS-CoV-2 proteomes. An initial analysis using a panel of sequence-based predictors suggested the presence of multiple aggregation-prone regions (APRs) in these proteomes and revealed a strong aggregation propensity in some SARS-CoV-2 proteins. We then studied the in vitro aggregation of predicted aggregation-prone SARS-CoV and SARS-CoV-2 proteins and protein regions, including the signal sequence peptide and fusion peptides 1 and 2 of the spike protein, a peptide from the NSP6 protein, and the ORF10 and NSP11 proteins. Our results show that these peptides and proteins can form amyloid aggregates. We used circular dichroism spectroscopy to reveal the presence of $\beta$-sheet rich cores in aggregates and X-ray diffraction and Raman spectroscopy to confirm the formation of amyloid structures. Furthermore, we demonstrated that SARS-CoV-2 NSP11 aggregates are toxic to mammalian cell cultures. These results motivate further studies about the possible role of aggregation of SARS proteins in protein misfolding diseases and other human conditions.

The COVID-19 pandemic, caused by the SARS-CoV-2 virus, has caused devastating loss of human lives and disruptions to the global economy[1,2]. Intense studies are ongoing to better understand the molecular mechanisms of SARS-CoV-2 pathogenesis to identify targets for therapeutic interventions.

In this study, given the widespread phenomenon of protein misfolding and aggregation[3,4], we investigate its manifestation and possible implications in the case of the SARS-CoV-2 proteome. Quite generally, there could be at least three different aspects of the interplay between the viral and host proteomes in terms of protein

aggregation: (i) the functional aggregation of viral proteins may help the virus to hijack the replication machinery of a host cell[5,6], (ii) the aberrant aggregation of viral proteins may represent an additional mechanism by which viruses damage the host cells[7,8], (iii) viral particles can trigger the misfolding and aggregation of host proteins, resulting in damage to the host organism[9].

Some viral proteins are known to form amyloid aggregates implicated in viral pathogenesis. An example is the protein PB1 of the influenza A virus, which forms one of the three subunits of the viral polymerase. During the early stages of viral infection, PB1 is expressed

[1]School of Biosciences and Bioengineering, Indian Institute of Technology Mandi, Kamand, Himachal Pradesh 175075, India. [2]Centre for Misfolding Diseases, Yusuf Hamied Department of Chemistry, University of Cambridge, Cambridge CB2 1EW, UK. [3]Structural Biology Laboratory, G. N. Ramachandran Protein Centre, CSIR-Institute of Microbial Technology, Chandigarh 160036, India. [4]Department of Medicinal Chemistry, Faculty of Ayurveda, Institute of Medical Sciences, Banaras Hindu University, Varanasi, Uttar Pradesh 221005, India. ✉e-mail: rajanishgiri@iitmandi.ac.in

in its monomeric form but then accumulates into amyloid forms at a later stage of infection, which are toxic to the infected cells[7,10,11]. Another example concerns the highly mutating H1N1 influenza A virus, whose nuclear export protein exhibits an intrinsic property to form aggregates, which is correlated with its role in virion budding[12,13]. Viral pathogenesis mediated by viral protein aggregates has also been shown for the protein M45 of murine cytomegalovirus[8]. We also note that viral capsid proteins may be prone to aberrant aggregation upon dysregulation of the functional self-assembly process, a process dependent on environmental conditions and proceeds via nucleation and growth[14].

It has been reported that in SARS-CoV, which is closely related to SARS-CoV-2, the membrane (M) protein, one of the membrane-forming proteins of the virus, can undergo aggregation[15]. It has also been shown that the C-terminal end of the envelope (E) protein of SARS-CoV includes an aggregation-prone motif and that this peptide can form aggregates in vitro[16]. Furthermore, the transmembrane domain (TMD) of protein E oligomerizes to form pentameric non-selective ion channels that might act as a viroporin, a small membrane-embedded protein having ion-conducting properties[17]. Also, in SARS-CoV, aggregated forms of ORF8b induce endoplasmic reticulum stress, lysosomal damage, and activation of autophagy[18].

Following these reports about the aggregation of coronavirus proteins, here we aim to investigate the aggregation propensity in the proteins of SARS-CoV and SARS-CoV-2. The RNA genome of SARS-CoV-2 encodes 29 proteins, which can be divided into the structural, accessory, and non-structural proteins (Fig. 1a, b)[19]. Our results identify most of the proteins which play crucial role in virus pathogenesis and survival, as aggregation-prone. We also compared the aggregation propensities of the SARS-CoV-2 proteins with those of the SARS-CoV proteins. We then focused our analysis on specific proteins by further investigating the Spike (S) protein fusion peptides 1 and 2, NSP11 of

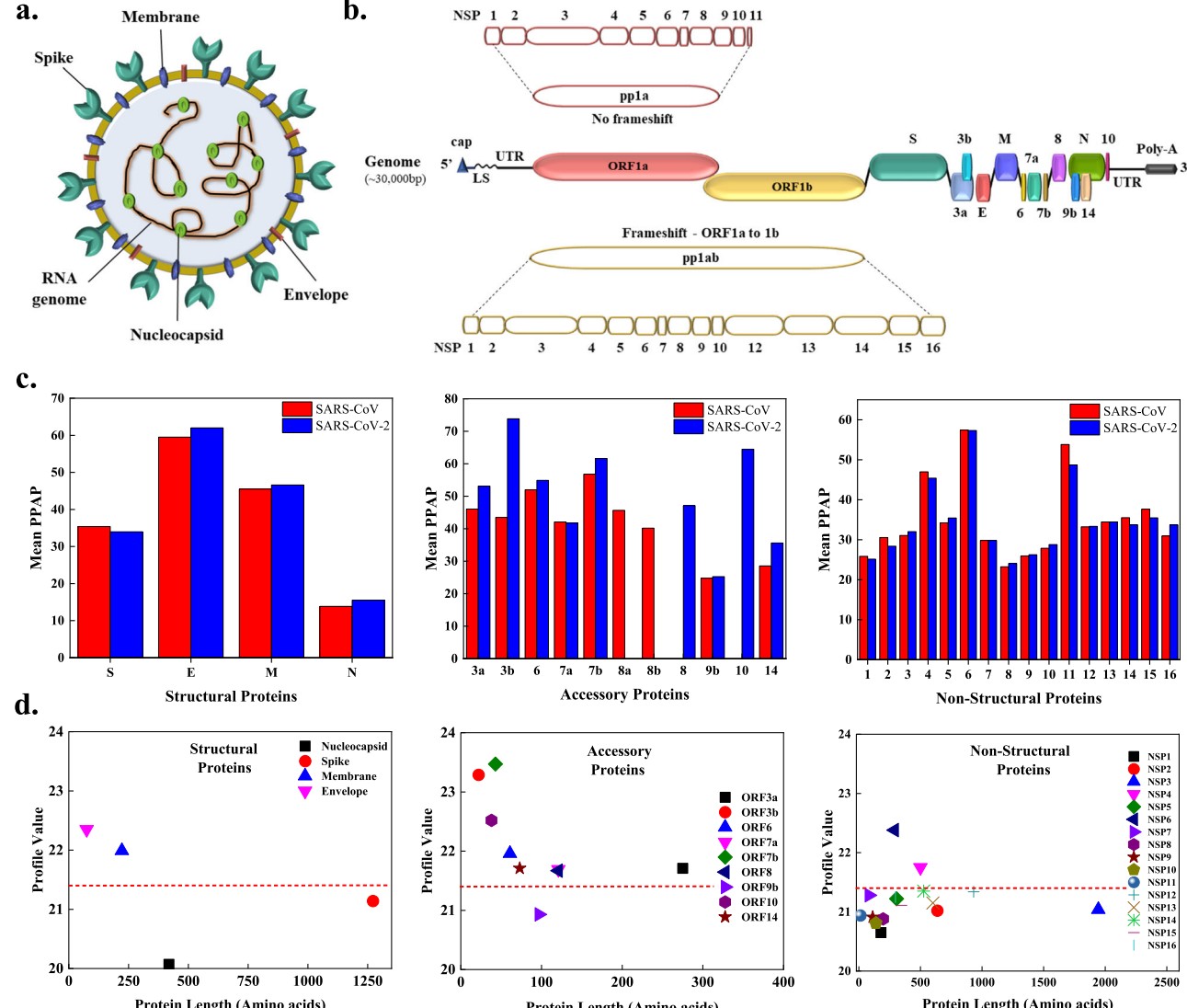

**Fig. 1 | Schematic illustration of the organization of genome and proteome of SARS-CoV-2 and aggregation propensity analysis of the SARS-CoV and SARS-CoV-2 proteomes. a** The SARS-CoV-2 viral particle comprises of positive-sense single-stranded RNA, which is associated with the nucleocapsid protein (N), and three surface proteins, spike (S), membrane (M), and envelope (E), embedded in the lipid bilayer. **b** The ~30 kbp long genome of SARS-CoV-2 encodes for four structural, nine accessory, and sixteen non-structural proteins (NSPs). **c** Comparison of mean predicted percentage aggregation propensity (PPAP) calculated using the mean percentage of APRs obtained from four servers (MetAmyl, AGGRESCAN, FoldAmyloid, FISH Amyloid) for SARS-CoV and SARS-CoV-2 structural, accessory, and non-structural proteins. **d** Average profile value for SARS-CoV-2 proteins obtained from FoldAmyloid analysis of proteins against protein length for structural, accessory, and non-structural proteins. The analysis was done at default settings in the FoldAmyloid server (threshold: 21.4, represented by the red-coloured short-dashed line and scale, i.e., the expected number of contacts within 8 Å).

SARS-CoV and signal sequence peptide, and fusion peptides 1 and 2 of the S protein, full-length ORF10, NSP6-p (residues 91–112 of NPS6), and NSP11 of SARS-CoV-2. For this purpose, we employed fluorescence, circular dichroism (CD) and Raman spectroscopies, and X-ray diffraction (XRD) methods to probe the formation of amyloid structures, as well as atomic force microscopy (AFM) and high-resolution transmission electron microscopy (HR-TEM) to visualize the morphology of resultant aggregates. In addition, we investigated the cytotoxicity of SARS-CoV-2 NSP11 aggregates on different mammalian cell lines.

## Results

### Aggregation-prone regions in the SARS-CoV and SARS-CoV-2 proteomes

The tendency to self-assemble into amyloid structures is an intrinsic property of proteins and depends on the presence of APRs within their amino acid sequences[20–22]. This tendency competes with that of self-assembling into functional complexes[23,24]. To investigate this competition, in this study, we analyzed the tendency of SARS-CoV and SARS-CoV-2 proteins to form amyloid aggregates using different prediction methods. We employed a combination of three different individual predictors, FISH Amyloid, AGGRESCAN, and FoldAmyloid, and a meta-predictor, MetAmyl, to analyse the presence of APRs. We also used CamSol to predict the solubility of proteins of both viruses. According to the results, APRs are abundant in both proteomes, and all proteins were found to contain at least one APR (Supplementary Tables 1–6).

For the comparison of amyloidogenic propensity of the SARS-CoV-2 and SARS-CoV proteomes, we calculated the mean predicted percentage amyloidogenic propensity (PPAP) from APRs obtained from the four predictors (MetAmyl, FISH Amyloid, AGGRESCAN, and FoldAmyloid) for both viruses (Fig. 1c). Numerous proteins, particularly the accessory proteins of SARS-CoV-2, were observed to be more amyloidogenic than the accessory proteins of SARS-CoV.

In SARS-CoV-2, among the structural proteins, membrane (M) and envelope (E) are found to be more amyloidogenic by FoldAmyloid in comparison with nucleocapsid (N) and spike (S) (Fig. 1d and Supplementary Table 1), and the accessory proteins to have more abundance of APRs (Supplementary Table 2). Except for ORF9b, all other proteins show FoldAmyloid profile values above the threshold (Fig. 1d and Supplementary Table 3). The 16 NSPs perform diverse roles, such as evading the host immune system, protection from host defense mechanisms, virus replication, and spreading the infection[19]. From the FoldAmyloid analysis, after plotting the average profile values for NSPs (Fig. 1d), NSP4 and NSP6 show values above the cut-off, which indicates a highly amyloidogenic nature.

To gain insights into the possible cleavage of APRs, 20S proteasome cleavage sites within the entire SARS-CoV-2 proteome were predicted by the NetChop 3.1 server (Supplementary Tables 7–9). In the context of our study, the reason for identifying these sites is twofold. First, the predicted sites inside APRs suggest that the proteasome may be unable to cleave the viral proteins due to aggregation. Secondly, the proteasome could cleave the viral proteins and release amyloidogenic regions inside the host cell. Here, in the case of accessory proteins, we found many cleavage sites in APRs (Supplementary Tables 7–9).

### Prediction of aggregation-prone regions in the structural proteins

Four structural proteins of SARS-CoV-2 participate in the virion assembly and packaging processes and in providing structure to the virus[25]. We analyzed the APRs of these structural proteins given in Supplementary Table 1. One of these proteins, S, is a heavily glycosylated transmembrane protein whose N-terminal S1 domain harbors receptor binding sites for the host cell. Its C-terminal S2 domain mediates the fusion between the virion and host cell membrane[26]. We predicted several APRs in the S protein (Supplementary Table 1).

We also predicted several APRs in the other structural proteins (M, E, and N). The membrane protein M gives shape to the virus, promotes viral membrane curvature, and binds with the nucleocapsid RNA complex during virus packaging. The envelope protein E is a transmembrane protein with an ion-channel activity that facilitates the assembly and release of viral particles. The amyloid-forming propensity of 9-residue stretch (TVYVYSRVK) of E has been reported previously for SARS-CoV[16]. Finally, the N protein, the proteinaceous part of the viral nucleocapsid, interacts with viral RNA and helps its packaging into the virion. According to our analysis, AGGRESCAN detected 18% amyloidogenic regions in SARS-CoV-2 N protein and only 16% in SARS-CoV N protein. Similar to these results, FoldAmyloid also predicted only 11% amyloidogenic regions in SARS-CoV N, ~3% less than SARS-CoV-2 N, which contains a total of ~14% amyloidogenic regions (Supplementary Tables 1 and 4).

### Prediction of aggregation-prone regions in the accessory proteins

The coronavirus genome codes for proteins termed accessory, which are multifunctional proteins that play an essential role in modulating the host response to the viral infection, such as downregulation of interferon pathways, release of proinflammatory cytokines and chemokines, and the induction of autophagy[19]. According to the predictors used in this study, all accessory proteins have multiple APRs (Supplementary Table 2). ORF3a and the N-terminal regions of ORF6, ORF7a, ORF7b, ORF8, and ORF10 have the potential to aggregate. In addition, the C-terminal regions of the ORF7a, ORF8, ORF9b, ORF10, and ORF14 proteins also exhibit numerous APRs.

### Prediction of aggregation-prone regions in the non-structural proteins

The SARS-CoV-2 proteome includes 16 non-structural proteins, all of which were found to comprise APRs in their sequences (Supplementary Table 3), which could play a role in their behaviours and interactions with the host cells. NSP1 creates a suitable viral propagation environment by blocking host cell translation and host immune response. NSP3 is a papain-like protease that cleaves the viral ORF1ab polyprotein at the NSP1/NSP2, NSP2/NSP3, and NSP3/NSP4 boundaries and interferes with the proper functioning of the host proteome by blocking the cellular degradation system. NSP4 and NSP6 are transmembrane proteins that may have a scaffolding activity for viral replication vesicle formation[27]. NSP5 is a serine-like protease that catalyzes the rest eleven cleavage events of the ORF1ab polyprotein. NSP12 is an RNA-dependent RNA polymerase, and NSP7 and NSP8 function as its processivity clamps. NSP10 is a cofactor for NSP16, which protects viral RNA from host antiviral measures. NSP13 is an RNA helicase, and NSP14 is a methyltransferase that adds a 5′ cap to viral RNA and is involved in proofreading the viral genome by virtue of its 3′–5′ exonuclease activity. NSP15 is an endoribonuclease that has a defensive role from host attacks[19].

### Experimental analysis of aggregation-prone SARS-CoV and SARS-CoV-2 proteins

To experimentally test the computational predictions of APRs in the SARS-CoV and SARS-CoV-2 proteomes, we investigated the in vitro aggregation behaviour of S protein fusion peptides 1 and 2 and NSP11 of SARS-CoV and the S signal peptide and fusion peptides 1 and 2, ORF10 protein, NSP6-p, and NSP11 protein of SARS-CoV-2. For this purpose, we selected physiological pH-temperature conditions and traced the aggregation process using the fluorescent dye thioflavin T (ThT). ThT gives a maximum emission peak at ~490 nm upon binding with β-sheets in amyloid fibrils[28,29]. Further, we studied the aggregation reactions using ThT fluorescence ($\lambda_{max}$ at 490 nm) in the presence of a fixed volume of incubated samples (25 μM)[30] (see Methods). We then utilized tapping-mode AFM and HR-TEM to gain insights into the

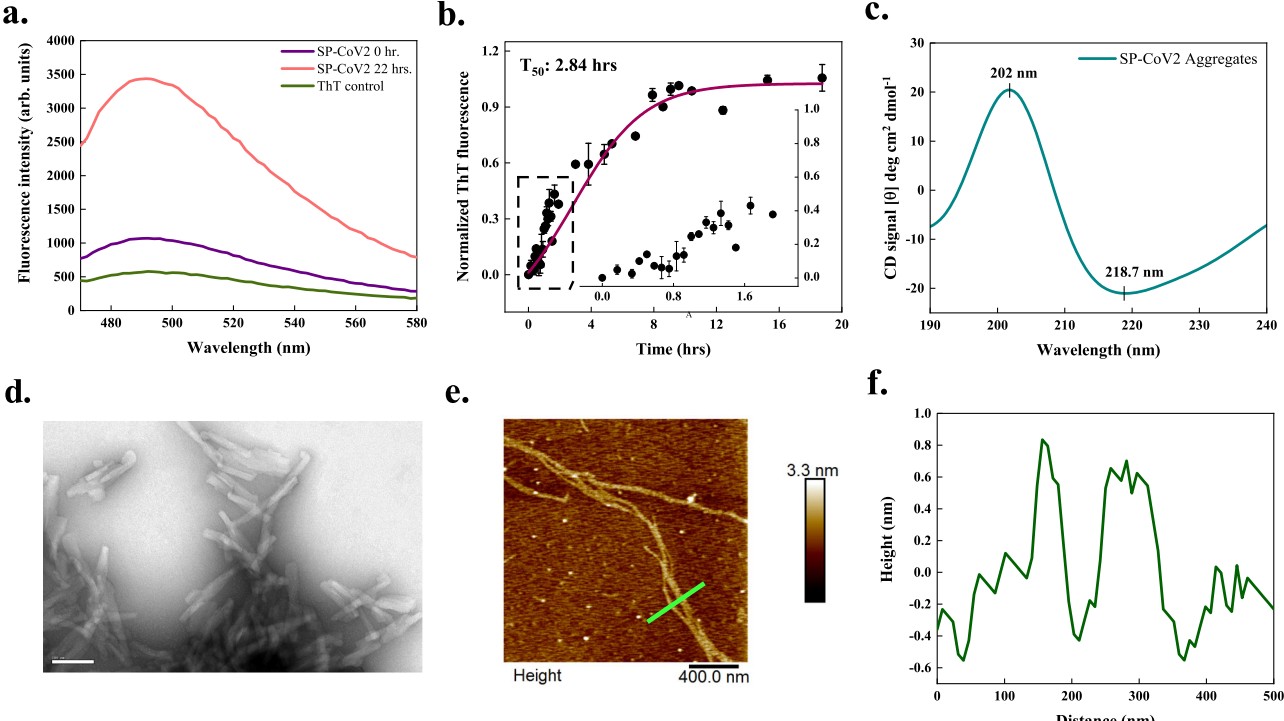

**Fig. 2 | In vitro aggregation of signal peptide (SP-CoV2) of SARS-CoV-2 S protein. a** ThT fluorescence scan ($\lambda_{ex}$ 440 nm) shows a ~3-fold increase at $\lambda_{max}$ in presence of SP-CoV2 at 22 h. **b** Aggregation kinetics monitored using ThT fluorescence at 490 nm. Each dotted symbol represents the average value of two technical replicates and error bar represents the SEM of these replicates. The wine-coloured line shows the sigmoidal fitting of data points that determines the $T_{50}$ of this reaction to be 2.84 ± 0.43 h. The aggregation kinetics shows a short lag phase that can be seen in the inset. **c** Secondary structural analysis of 0.25 mg/ml aggregates (240 h) using CD spectroscopy reveals two peaks having positive (202 nm) and negative ellipticities (218.7 nm), corresponding to the presence of $\beta$-sheet structures. The spectrum is smoothened with 10 points using the FFT filter function. **d** HR-TEM imaging reveals twisted fibrillar structures (100 nm scale bar) at 240 h (10 micrographs were captured from the same grid). **e** After 170 h incubation, SP-CoV2 fibrils appear branched through AFM (scale bar 400 nm). Seven micrographs were captured from the same sample. **f** Height profile of two fibrils shown with a green-coloured line in panel (**e**). Additional micrographs are given in Supplementary Fig. 2. The source data are given in the Source Data file. Arb. units are arbitrary units.

morphological features of the aggregates. The maturation of fibrils results in structural rearrangement and enhancement in $\beta$-sheet rich structures. Therefore, for studying the secondary structure composition, we employed CD spectroscopy. We also performed XRD and Raman spectroscopy to obtain evidence for the formation of amyloid structures by selected peptides and proteins. Furthermore, to examine the seeding activity of aggregates, we performed the self-seeding experiment with aggregates of S fusion peptide 2 of SARS-CoV (Supplementary Fig. 1).

## Signal peptide of the S protein

The S protein plays a vital role in receptor recognition and cell membrane fusion. Its mature form contains four regions, a signal sequence, an ectodomain, a transmembrane domain, and an endodomain, further divided into the S1 and S2 subunits. The 12-residue N-terminal signal sequence (SP-CoV2) directs the S protein to its destination in a viral membrane[31]. CamSol predicted this peptide to be poorly soluble, and aggregation prediction servers identified it as an APR (Supplementary Table 1). Our in vitro study of the aggregation behaviour of SP-CoV2 (see Methods) showed an over 3-fold increase in ThT fluorescence intensity upon aggregation (Fig. 2a). The nucleation process exhibited a very short lag phase, suggesting that the formation of the initial aggregates takes place rapidly during the incubation time, as expected for a peptide of poor solubility, and reached saturation with a half-time ($T_{50}$) of about 2.84 h (Fig. 2b). Figure 2c depicts the spectrum obtained using CD spectroscopy. A positive peak at 202 nm and a negative peak near 218 nm provided evidence for the $\beta$-sheet rich structures in amyloids formed by SP-CoV2. Further, HR-TEM imaging

of SP-CoV2 aggregates displayed short but slender fibrils formed after 240 h (Fig. 2d). Similar overlapping fibrils are observed under AFM after 170 h (Fig. 2e, f).

## Fusion peptides 1 and 2 of the S proteins

The S protein contains two fusion peptides (FP1 and FP2) of approximately 22 amino acids in the S2 subunit that helps the virus penetrate the host cell membrane. These regions display calcium-dependent membrane-ordering properties to trigger membrane fusion[31,32]. All the predictors used in this study identified APRs in both fusion peptide regions of the S proteins. We have analysed the aggregation in these regions in vitro, finding increments in their ThT fluorescence intensities (Figs. 3 and 4).

In the case of FP1 of SARS-CoV (FP1-CoV), a ~3.5-fold increase in ThT fluorescence is observed after incubation of 125 h at 1000 rpm at 37 °C (Fig. 3a). The kinetics of aggregation showed a $T_{50}$ of 16.44 h with a lag phase of about 11.44 h (Fig. 3b). Secondary structures of monomeric and aggregated FP1-CoV peptide studied using CD revealed major structural rearrangements (Fig. 3c). The monomeric peptide showed a typical CD spectrum of disordered conformation, while the aggregated sample had a CD spectrum corresponding to β-sheets. This indicates the transition from disordered to $\beta$-sheet containing structures of the fibrils. Images captured using HR-TEM in Fig. 3d depicts slender and branched fibrils with twisted morphology, a characteristic of amyloids. Tapping-mode AFM also revealed analogous amyloid fibrils of FP1-CoV with a significant height of 6–8 nm (Fig. 3e, f). Additional TEM and AFM micrographs are given in Supplementary Fig. 2.

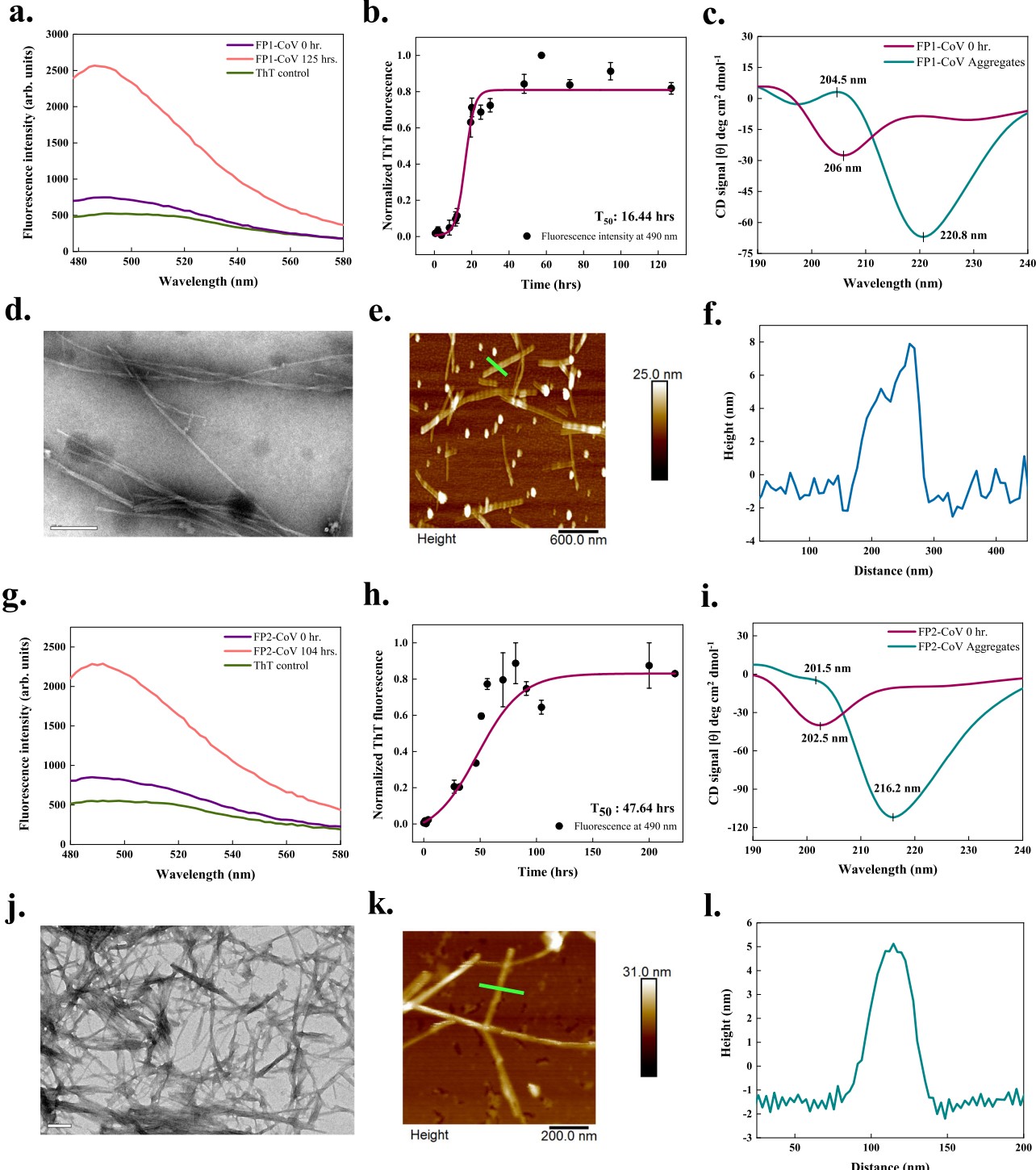

**Fig. 3 | In vitro aggregation of fusion peptides 1 (FP1-CoV) and 2 (FP2-CoV) of SARS-CoV S protein. a, g** ThT fluorescence scan ($\lambda_{ex}$ 440 nm) shows ~3.5- and ~3-fold increment at $\lambda_{max}$ with FP1-CoV and FP2-CoV incubated samples, respectively. **b, h** Aggregation kinetics monitored using ThT fluorescence. Each dotted symbol represents the average value of two (FP2-CoV) and three (FP1-CoV) technical replicates and error bar represents the SEM of these replicates. The wine-coloured lines show the sigmoidal fittings of data points that determines the $T_{50}$ values as 16.44 ± 0.84 h and 47.64 ± 3.75 h for FP1-CoV and FP2-CoV peptides, respectively. **c, i** The secondary structure of 1 mg/ml monomeric and aggregated (240 h) peptides are investigated using CD spectroscopy. The spectra of the monomeric FP1-CoV show a negative peak at 206 nm, exposing a molten globule-like structure. For aggregates, two characteristic peaks, one positive at 204.5 nm and a negative peak at ~220 nm, show the presence of $\beta$-sheet rich secondary structure. Spectral data of FP2-CoV reveal disordered structure for monomer and $\beta$-sheet structure for aggregates. The spectra are smoothened with 10 points using FFT filter function. **d, j** FP1-CoV fibrils at 240 h appear to be longer, slender and twisted under HR-TEM. Whereas image of FP2-CoV aggregates (104 h) show highly branched and pointed fibrils; scale bar represents 100 nm and 200 nm for images shown in panels (**d**) and (**j**), respectively (11 and 17 TEM micrographs were captured for FP1-CoV and FP2-CoV peptides, respectively). (**e, k**) AFM imaging of FP1-CoV and FP2-CoV aggregates at 240 and 270 h; scale bars represent 600 nm and 200 nm. Four and 7 AFM micrographs were captured for FP1-CoV and FP2-CoV aggregated samples, respectively. Due to the double-tip effect during scanning, the imaged fibrils were blurry in panel (**e**). **f, l** Height profiles of FP1-CoV and FP2-CoV fibrils shown with green-coloured lines in panels (**e**) and (**k**), respectively. The source data are given in the Source Data file. Arb. units are arbitrary units.

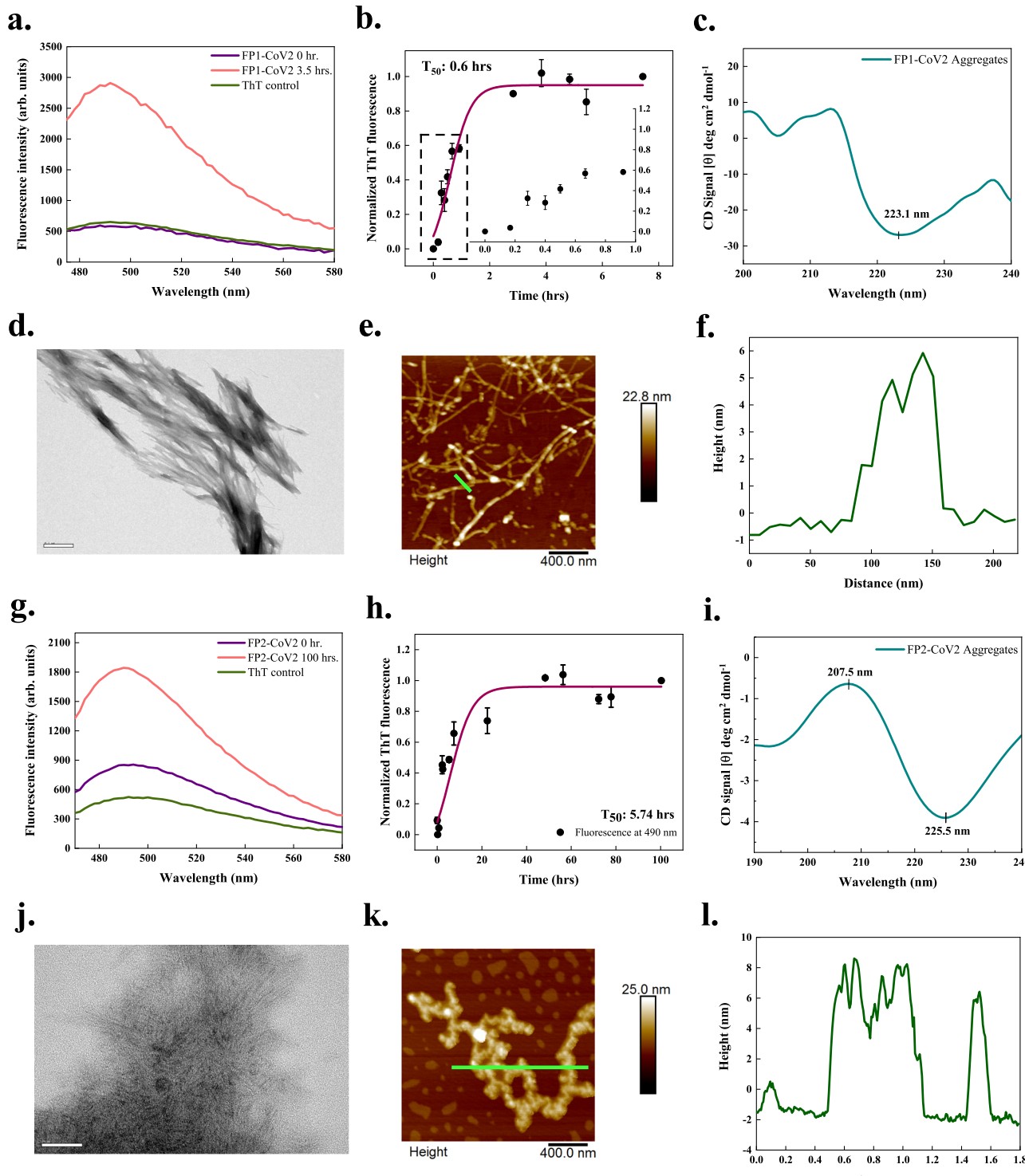

**Fig. 4 | In vitro aggregation of fusion peptides 1 (FP1-CoV2) and 2 (FP2-CoV2) of the SARS-CoV-2 S protein. a, g** ThT fluorescence scan ($\lambda_{ex}$ 440 nm) shows a ~6- and ~2-fold increment at $\lambda_{max}$ with FP1-CoV2 and FP2-CoV2 incubated samples, respectively. **b, h** Aggregation kinetics monitored using ThT fluorescence. Each dotted symbol represents the average value of two technical replicates and error bar represents the SEM of these replicates. The wine-coloured lines show the sigmoidal fittings of data points that determines the $T_{50}$ values as 0.6 ± NC h and 5.74 ± NC h for FP1-CoV2 and FP2-CoV2 peptides, respectively. The kinetics reaction of FP1-CoV2 shows a very short lag phase that can be seen in the inset. **c, i** The secondary structure of 1 mg/ml peptides (both incubated for 240 h) investigated using CD spectroscopy shows a negative peak at ~223 and ~225 nm. The spectral data of FP1-CoV2 and FP2-CoV2 aggregates are smoothened with 4 and 10 points using the FFT filter function. **d, j** HR-TEM imaging of FP1-CoV2 (96 h) and FP2-CoV2 (100 h) exposed branched and long fibrils; scale bar represents 500 nm and 50 nm for images in panels (**d**) and (**j**). Sixteen and 22 TEM micrographs were captured for FP1-CoV2 and FP2-CoV2 peptides, respectively. **e, k** AFM images of FP1-CoV2 (96 h) and FP2-CoV2 (100 h) also reveal highly branched fibrils; scale bars represent 400 nm for both images. Seven and 5 AFM micrographs were captured for FP1-CoV2 and FP2-CoV2 aggregates, respectively. **f, l** Height profiles of FP1-CoV2 and FP2-CoV2 fibrils shown with green-coloured lines in panels (**e**) and (**k**), respectively. Having a height of ~8 nm, FP2-CoV2 amyloids are slightly bigger. Additional TEM and AFM micrographs are given in Supplementary Fig. 2. NC stands for not calculated. The source data are given in the Source Data file. Arb. units are arbitrary units.

Upon aggregation of FP2 of SARS-CoV (FP2-CoV), we found an increase of ~3-fold in ThT fluorescence after 104 h of constant stirring at 1000 rpm and incubation at 37 °C (Fig. 3g). The aggregation kinetics followed by FP2-CoV is slower than FP1-CoV, where the $T_{50}$ is measured to be about 47.64 h and $T_{lag}$ of 8.44 h, as shown in Fig. 3h. Far-UV CD spectra of monomer and aggregated peptide illustrated in Fig. 3i clearly exhibit the differences in secondary structures. Monomers in disordered states transformed into β-sheet structures as displayed by the shift in negative ellipticity from peaking at 202.5 nm for monomers and 216 nm for aggregates. Further, another peak at 201.5 nm can be seen in the spectra of aggregates, but due to the high negative signal at 216 nm, it overall has a negative ellipticity. HR-TEM images exposed remarkably pointed and highly-branched fibrillar structures of amyloids (Fig. 3j). Furthermore, the amyloids have a height of ~5 nm, as visualized using tapping-mode AFM (Fig. 3k, l). Additional micrographs of FP2-CoV aggregates are given in Supplementary Fig. 2.

Upon a 3.5 h incubation of FP1 of SARS-CoV-2 (FP1-CoV2), we found an increased ThT fluorescence by ~6-fold compared to the monomeric peptide. The studied aggregation reaction has a $T_{50}$ value of about 0.6 h (Fig. 4a, b). Far-UV CD spectrum of FP1-CoV2 aggregates displayed a negative peak at ~223 nm (Fig. 4c). This shows the development of a secondary structure content after aggregation. The positive peak near ~200 nm, also corresponding to β-sheet, is unclear due to the access noise. Furthermore, at 96 h, numerous entangled and significantly branched filaments of FP1-CoV2 were seen after TEM imaging (Fig. 4d). AFM images illustrated in Fig. 4e also presented identical fibrils after 96 h of incubation. These differently sized filamentous aggregates have a height distribution peaking at ~6 nm (Fig. 4f).

In vitro aggregation of FP2 of SARS-CoV-2 (FP2-CoV2), as shown in Fig. 4g, the ThT fluorescence intensity showed a change of ~2-fold after incubation. Furthermore, the $T_{50}$ of its aggregation kinetics is revealed to be about 5.74 h (Fig. 4h). Akin to FP1-CoV2 aggregates, FP2-CoV2 amyloids also showed a negative CD signal at ~225 nm. However, due to the low signal, the peak near 207 nm has overall negative ellipticity (Fig. 4i). Images obtained using TEM revealed typically branched and overlapping aggregates in Fig. 4j. Further, the FP2-CoV2 aggregates, upon 100 h incubation, appeared branched under AFM, having a slightly longer height peaking at ~8 nm (Fig. 4k, l).

The results of kinetics experiments of FP1 and FP2 peptides from both viruses imply that FP2 peptides follow slower aggregation reactions than the FP1 peptides. This can be rationalised based on their amino acid compositions. The FP1 peptide is composed mainly of aggregation-deriving hydrophobic residues such as alanine, leucine, isoleucine, methionine, and phenylalanine. In contrast, the FP2 peptide consists of glutamine, cysteine, lysine, the aspartic acid, in addition to hydrophobic residues like alanine, leucine, and isoleucine. Also, fusion peptides of SARS-CoV-2 have faster aggregation reactions than their SARS-CoV counterparts, which can also be correlated to the substitutions that have occurred in the SARS-CoV-2 S protein.

## ORF10 protein (full-length)
The SARS-CoV-2 ORF10 translates into a 38-residue long protein that does not have significant homology with any known proteins. Although the evidence of the presence of ORF10 in SARS-CoV-2 infected cells is limited[33,34], it has been investigated to have a high dN/dS value (non-synonymous over synonymous substitution rate; 3.82)[35]. We predicted ORF10 to contain multiple APRs (Supplementary Table 2). In vitro aggregation experiments, ORF10 showed ~7-fold increase in ThT fluorescence in comparison with the freshly dissolved monomer (Fig. 5a). The $T_{50}$ of aggregation reaction is slightly longer, about 27.11 h considering the presence of a very short lag phase (Fig. 5b). Secondary structure analysis using the CD spectrum showed a negative peak at 228.7 nm. Another peak having positive ellipticity

near 203 nm is suggestive of the presence of β-sheet structures in aggregates (Fig. 5c). Further, the formation of amorphous aggregates by ORF10 protein is revealed by performing morphological analysis using AFM and HR-TEM. Compared with fibrillar aggregates of other peptides studied in this report, ORF10 protein forms amorphous aggregates (Fig. 5d, e and Supplementary Fig. 3). The ThT fluorescence of ORF10 protein aggregates was maintained at saturation while preparing the samples for AFM and TEM imaging.

## NSP6-p
The SARS-CoV-2 NSP6 protein is an essential host immune system antagonist where it antagonizes type I interferon by blocking TANK binding kinase[36]. While accommodating multiple transmembrane regions, residues 91–112 of NSP6 are particularly important since they lie outside the membrane and can interact with host proteins[37]. This region exhibits APRs as predicted using all servers (Supplementary Table 3). The intrinsic solubility of this region is also calculated to be low by CamSol. Its aggregation using ThT fluorescence is detected after 102 h of incubation at 1000 rpm at 37 °C (Fig. 6a). Further, the kinetic analysis revealed a 16.64 h long lag phase followed by an exponential increase before the final plateau phase was reached (Fig. 6b). The $T_{50}$ value of aggregation kinetics of NSP6-p is calculated to be 21.66 h. Furthermore, HR-TEM imaging in Fig. 6c exposed fibrils' overlapping and twisted nature after 240 h incubation. Figure 6d, e shows the AFM images of a 96 h aggregated sample where the height profile revealed a peak height of ~3 nm (Fig. 6f). Additional TEM and AFM micrographs of NSP6-p aggregates are given in Supplementary Fig. 3.

## NSP11
NSP11 of SARS-CoV (NSP11-CoV) is a 13-residue long peptide and only differs from NSP11-CoV2 at positions 5th and 6th[19]. FISH Amyloid and AGGRESCAN predicted regions 6–10 and 6–13 in this protein, similar to NSP11-CoV2. MetAmyl predicted a more extended region of 4–13 residues in this protein (Supplementary Table 6). In vitro, an increment in ThT fluorescence of 2-fold is observed after 143 h (Fig. 7a). The secondary structure determined using CD showed a signature peak at ~200 nm for monomeric peptide, representing its disordered structure. Upon incubation, the negative peak of spectra shifts from 200 to 217 nm which shows the conformational change to β-sheet secondary structure in aggregates (Fig. 7b). However, no positive CD signal near 200 nm is observed in aggregates. Further, tapping-mode AFM images revealed longer and slender NSP11-CoV fibrils with a height of ~2–3 nm (Fig. 7c and Supplementary Fig. 4).

The first 9 residues of 13-amino acid SARS-CoV-2 NSP11 (NSP11-CoV2) are similar to the N-terminal 1–9 residues of NSP12 (RdRp)[19]. AGGRESCAN predicted the region from 6–13, FISH Amyloid predicted residues 6–10, and MetAmyl predicted residues 8–13 in NSP11-CoV2 as APRs (Supplementary Table 3). There is only a ~2-fold increment in ThT fluorescence intensity on binding to the 192 h incubated sample compared to the monomeric peptide (Fig. 7d). According to the kinetic analysis of NSP11-CoV2 aggregation, following a lag phase of 41.11 h, the process reached a plateau phase with $T_{50}$ of about 66.73 h (Fig. 7e). The changes in secondary structure monitored using far-UV CD-spectroscopy are illustrated in Fig. 7f, where the spectrum of monomeric NSP11-CoV2 is representative of the disordered proteins, which is also reported in our previous study[38]. For aggregated protein, the CD spectra represent a robust negative peak near 218 nm and a positive peak at 198.5 nm, suggesting the presence of β-sheet secondary structure elements in aggregates. These results indicate that, upon aggregation, NSP11-CoV2 converts from disordered to a conformation with a considerable secondary structure. Further, using AFM and HR-TEM, typical amyloid fibrils (at 192 h) of ~2.5 nm height are detected (Fig. 7g–i).

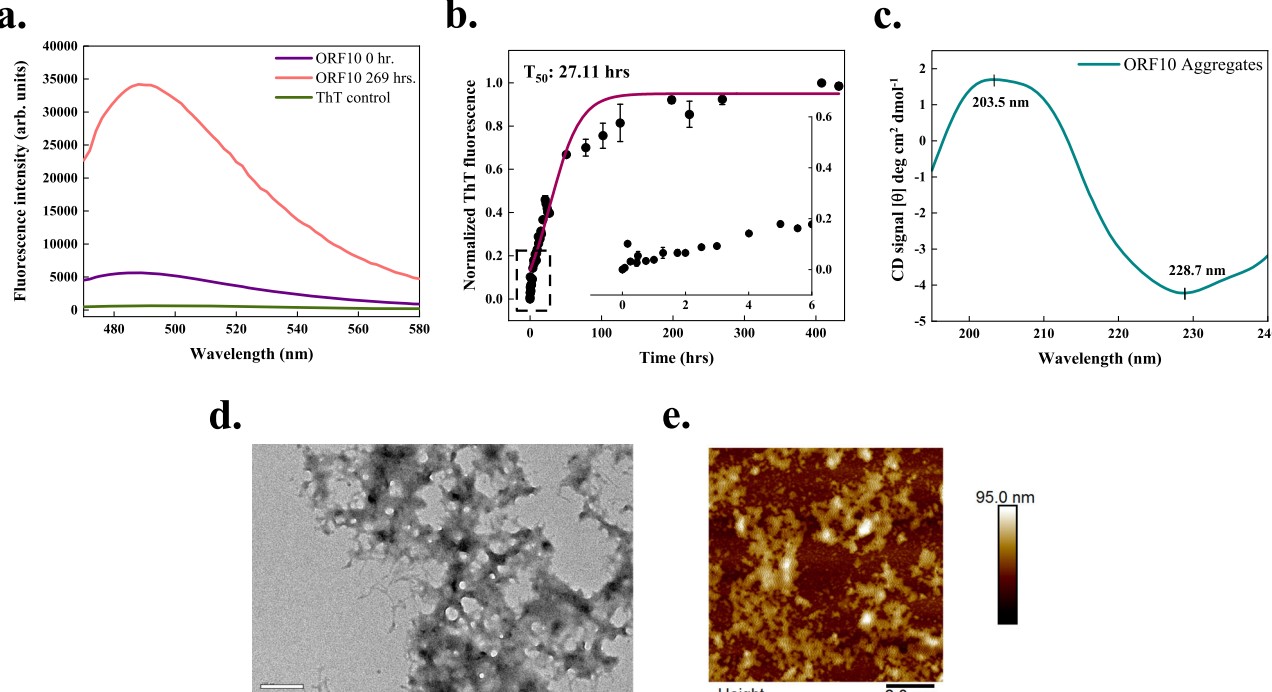

**Fig. 5 | In vitro aggregation of ORF10 protein of SARS-CoV-2. a** ThT fluorescence scan ($\lambda_{ex}$ 440 nm) indicates an increase of ~7-fold at $\lambda_{max}$ with ORF10 incubated sample. **b** Kinetics of ORF10 aggregation monitored using ThT fluorescence. Each dotted symbol represents the average value of two technical replicates and error bar represents the SEM of these replicates. The wine-coloured line shows the sigmoidal fitting of data points that determines the $T_{50}$ of this reaction to be 27.11 ± NC hrs. The aggregation kinetics shows a short lag phase that can be seen in the inset. **c** The secondary structure of 1 mg/ml incubated sample (240 h) studied using CD shows the presence of a positive and a negative peak. The data is smoothened with 10 points using the FFT filter function. **d** HR-TEM imaging of 96 h incubated sample revealed the amorphous nature of aggregates formed by ORF10 protein; the scale bar represents 200 nm. Eight micrographs were captured from the same grid. **e** Morphology of aggregates after 96 h incubation also confirms the amorphous type of aggregates under tapping-mode AFM; scale bar represent 2 µm. Eight micrographs were captured from the same sample. NC stands for not calculated. The source data are given in the Source Data file. Arb. units are arbitrary units.

## Analysis of protein aggregates using Raman spectroscopy

Raman spectroscopy enables the quantification of β-sheet content in aggregates[39,40]. This non-destructive technique does not require any treatment or external probes. Raman spectra of proteins consist of amide I–VII bands. The characteristic amide I–III bands occur due to the stretching of C=O bond of the peptide backbone, coupled C−N stretch and C−H bend, and out-of-phase stretching of C−N bond and bending of C−H bond, respectively[41]. Intensity changes and shifts in the positioning of these amide bands of monomeric and aggregated proteins can be linked to their structural differences[39,40]. Here, we collected the Raman spectra of monomers and aggregates of selected peptides to understand their secondary structures. The shifts in their amide I–III bands and vibrations assigned to the bending of the Cα−H bond are listed in Supplementary Table 10.

In the case of SP-CoV2, we observed significant changes in the amide II band. In monomeric peptide, the vibrational peak at 1548.3 cm⁻¹, representative of the disordered structure, showed a significant shift to 1554.2 cm⁻¹ for β-sheet. Amide I and III band positions also showed a slight shift, reflecting a gain in β-sheet rich structures (Fig. 8a). When FP1-CoV is aggregated, a narrow peak at 1671.2 cm⁻¹ of amid I is found to be shifted from 1663 cm⁻¹ in the monomer. Furthermore, ~2.5 cm⁻¹ shift in amide II vibrations is also suggestive of structural rearrangements from unstructured to β-sheet secondary structure (Supplementary Fig. 5). After FP2-CoV peptide fibril formation, narrowing of amide I vibrational peak at 1672.2 cm⁻¹ in addition to increased intensity is observed. A relatively higher intensity band of amide III of aggregates at 1236.9 cm⁻¹ showed a shift of ~13 cm⁻¹ from 1250 cm⁻¹ in the monomer. Further, the band associated with side-chain vibrations became lightly wide in aggregates peaking at 1445.5 cm⁻¹ from monomers, where the

vibrations peaked at 1439.3 cm⁻¹ (Fig. 8b). Upon aggregation of FP1-CoV2, a high peak of amide I at 1672.7 cm⁻¹ marking the presence of β-sheet, showed a shift from 1657.4 cm⁻¹ in the monomer. A slight shift in the amide II band causing vibrations is also found. The Amide III band of the aggregated peptide at 1235 cm⁻¹ indicated the β-sheet rich structure; however, the corresponding band in monomer is unclear (Fig. 8c). A clear shift in the amide I band from 1655.8 cm⁻¹ in monomer to 1673.1 cm⁻¹ in aggregates of FP2-CoV2 peptide is detected. Amide II band position is also observed to be shifted from 1554 cm⁻¹ in monomer to 1552.2 cm⁻¹ in aggregates, reflecting the changes in conformation. Likewise, bands arising from Cα − H bending occurred at 1385.5 cm⁻¹ and 1396 cm⁻¹ in monomer and aggregated peptides, respectively (Fig. 8d). The Raman spectral data of the ORF10 protein revealed shifts in amide I and II bands. Notably, a shift in side-chain vibrations from 1446.5 cm⁻¹ in monomer to 1464.3 cm⁻¹ in aggregated protein is observed. However, the amide III band in the ORF10 monomer is not precisely identified to study the shift in the spectra (Fig. 8e).

Overall, we observed shifts in Raman spectral properties in all the samples. Bands developing due to side-chain vibrations and Cα−H bond bending also showed significant shifts (Supplementary Table 10). In conclusion, Raman spectral data indicate the structural differences in monomers and aggregates.

## X-ray diffraction pattern (XRD) of aggregates

Despite having different native structures, sizes, and functions, amyloidogenic proteins can form amyloid aggregates structurally organized in a similar manner[3]. Consequently, amyloid fibrils show similar characteristic diffraction patterns on exposure to X-rays. Typically, XRD patterns of aligned mature fibrils exhibit a meridional reflection at

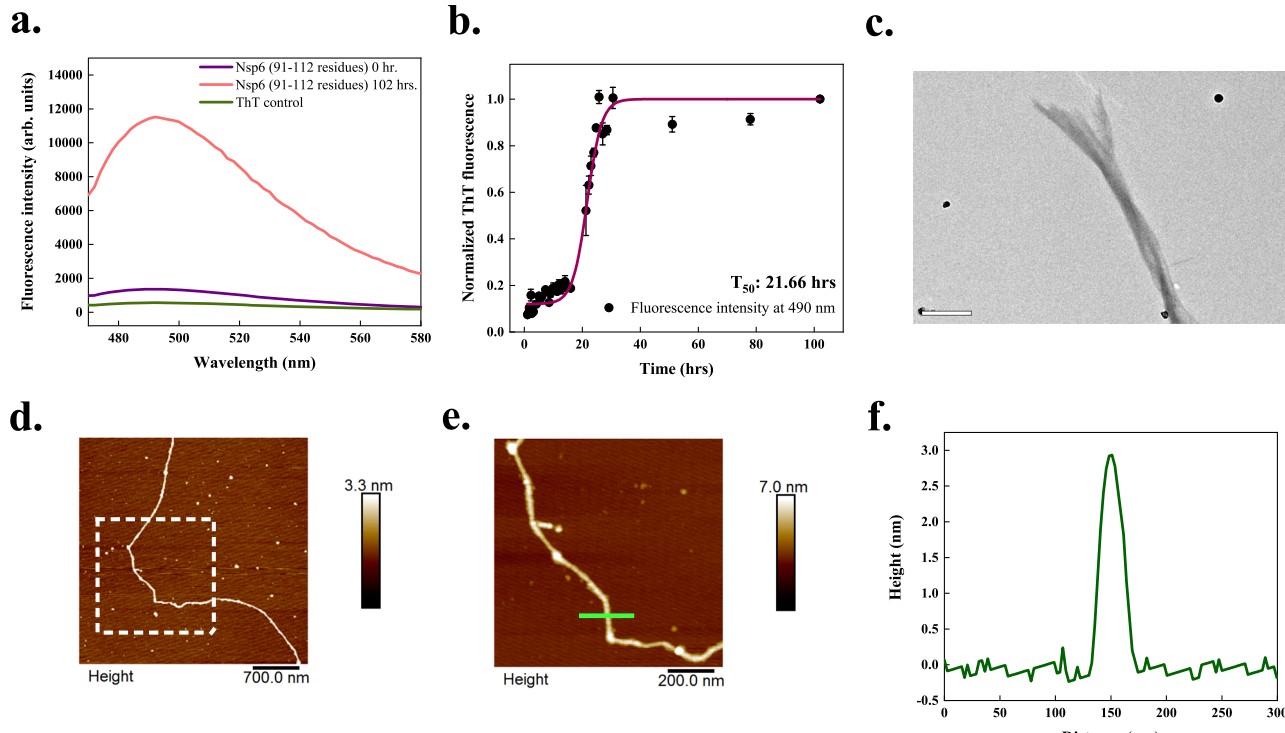

**Fig. 6 | In vitro aggregation of NSP6-p of SARS-CoV-2. a** ThT fluorescence scan ($\lambda_{ex}$ 440 nm) indicates an increase of ~10-fold at $\lambda_{max}$ with NSP6-p incubated sample. **b** Aggregation kinetics of NSP6-p monitored using ThT fluorescence. Each dotted symbol represents the average value of two technical replicates and error bar represents the SEM of these replicates. The wine-coloured line shows the sigmoidal fitting of data points that determines the $T_{50}$ of this reaction to be 21.66 ± 0.8 h. **c** Under HR-TEM, 240 h incubated aggregates are longer and branched, possessing twisted morphology; the scale bar represents 500 nm. Four micrographs were captured from the same grid. **d** AFM image of NSP6-p fibrils at 96 h with scale bar 700 nm. **e** A zoomed-in image of a fibril shown in a white box in panel (**d**) with scale bar 200 nm. A total of 5 micrographs were captured from the same sample. **f** The height profile of NSP6-p fibril is indicated with a green-coloured line in panel (**e**). CD spectrum of NSP6-p aggregates is not shown as it produced high noise rendering the spectral data unreliable. The source data are given in the Source Data file. Arb. units are arbitrary units.

around ~4.8 Å and an equatorial reflection between 10 and 12 Å. The meridian diffraction signal corresponds to the distance between $\beta$-strands, and a relatively more diffused equatorial signal arises from the inter-sheet network[42,43].

Here, we obtained XRD patterns of some aggregates whose aggregation properties are reported experimentally in previous sections. Non-oriented amyloid samples were subjected to X-ray fiber diffraction analysis. Aggregates formed by SP-CoV2, FP1-CoV, FP2-CoV, FP1-CoV2, and NSP6-p exhibit a classic meridional XRD reflection at ~4.7 Å (Fig. 9a–e). This kind of diffraction pattern confirms the alignment of $\beta$-strands in the fibrils with a spacing of ~4.7 Å. Furthermore, SP-CoV2, FP1-CoV, and NSP6-p display diffraction rings at 11 Å, ~10.2 Å, and 11.6 Å, respectively. These signals affirm the varying inter-sheet distance formed due to cross $\beta$-sheet structures in aggregates of these peptides. The diffraction pattern of SP-CoV2 and FP1-CoV also demonstrates rings at 21 Å and 16.7 Å, respectively, possibly corresponding to higher-order structures of fibrils. The XRD data presented here confirm the amyloid formation by these peptides. However, we did not observe a prominent diffraction ring at ~10 Å in FP2-CoV and FP1-CoV2 samples. Further, for ORF10 aggregates, no diffraction pattern is observed.

**Toxic effects of aggregates on the viability of mammalian cells**
To investigate whether the aggregation of the SARS-CoV-2 peptides reported above could be associated with host cell damage, we investigated whether or not $\beta$-sheet rich amyloid fibrils of NSP11-CoV2 are cytotoxic. We performed an MTT assay, a colorimetric-based cell viability test, to assess the effect of aggregates on the SH-SY5Y and HepG2

cell lines. The cells were treated separately with varying concentrations of NSP11-CoV2 monomers (used as control) and amyloid aggregates (192 h). No significant cell death is observed after 24 h (Fig. 10a, c). However, when treatment is extended to 72 h, cell death observed at the highest aggregate concentration is comparatively enhanced (Fig. 10b, d). We also observed that the percent cell viability in HepG2 cells is significantly reduced ($p < 0.0001$) more than in SH-SY5Y cells ($p < 0.01$), suggesting that the aggregates are comparatively more toxic to the liver cell line. HepG2 cell viability remained around 74%, while in SH-SY5Y, the percent cell viability remained around 86%. These results thus suggest that NSP11-CoV2 aggregates could be toxic to mammalian cells at relatively higher concentrations.

## Discussion
This study was inspired by a series of recent reports where brain disorders were correlated with the presence of viruses such as HIV, HSV-1, and H5N1[44–47]. The SARS-CoV-2 infection has been associated with severe neurological manifestations, including acute cerebrovascular diseases, skeletal muscle injury, acute hemorrhagic necrotizing encephalopathy, meningitis, and meningoencephalitis[48–51]. In the case of protein misfolding disorders, including Alzheimer's and Parkinson's diseases, SARS-CoV-2 is also reported to disrupt the homeostasis of tau and A$\beta$42[52–54]. A growing body of evidence also reveals the interaction of the SARS-CoV-2 N and S proteins with A$\beta$42, and $\alpha$-synuclein[55,56]. The S protein can upregulate the expression of $\alpha$-synuclein and accelerate its aggregation[55]. Its S1 subunit binds with high affinity with A$\beta$42 and dampens its clearance in serum[56]. In addition, the N protein is also shown to expedite the aggregation kinetics of $\alpha$-synuclein and disturb

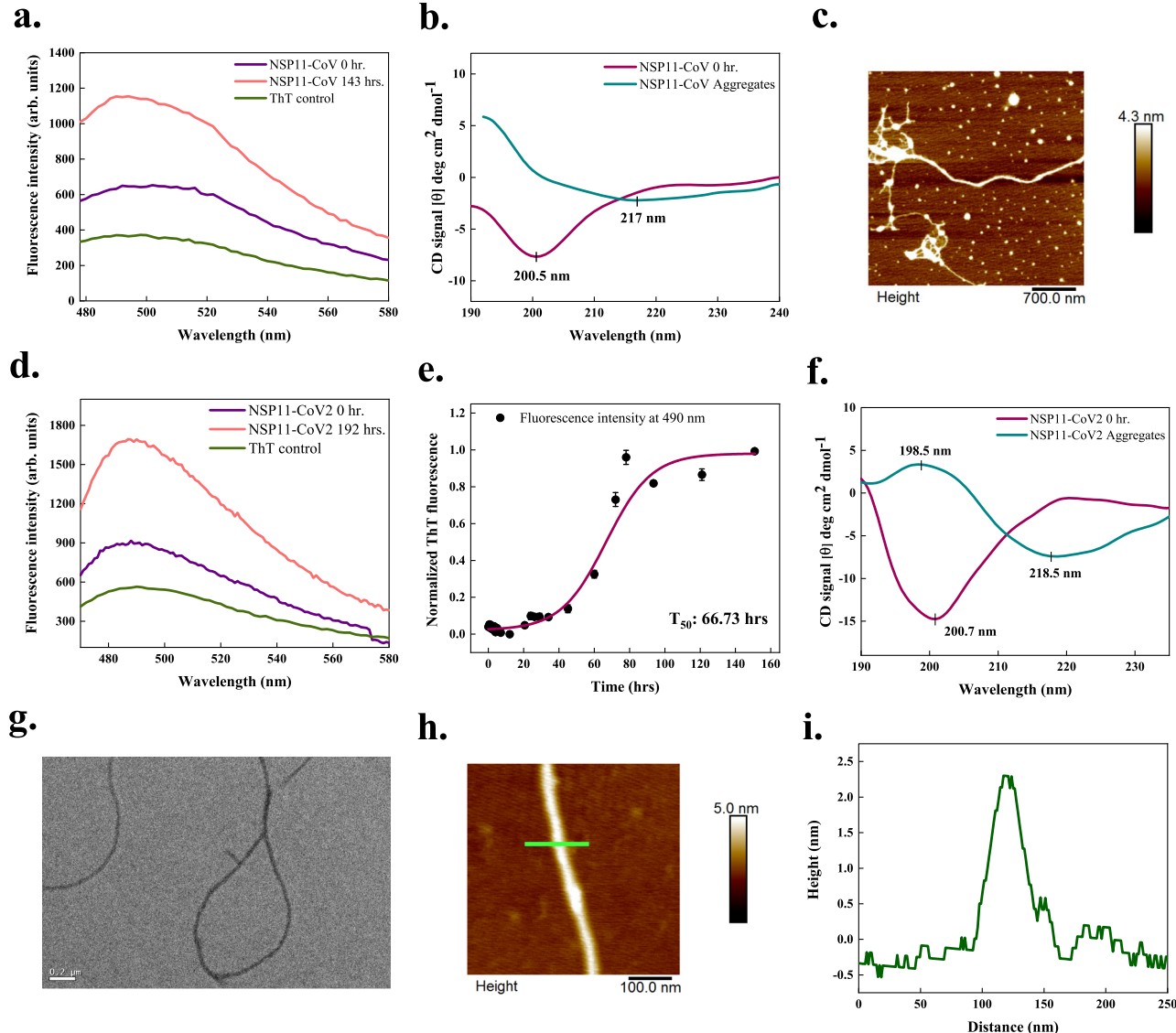

**Fig. 7 | In vitro aggregation of the NSP11 proteins of SARS-CoV (NSP11-CoV) and SARS-CoV-2 (NSP11-CoV2). a, d** ThT fluorescence scan ($\lambda_{ex}$ 440 nm) indicates an increase of ~2-fold at $\lambda_{max}$ with NSP11-CoV and NSP11-CoV2 incubated samples, respectively. **b, f** CD spectra of 3 mg/ml monomeric peptides show the signature peaks at ~200 nm for disordered secondary structures. For NSP11-CoV aggregates (after 120 days incubation), a negative peak at 217 nm is observed corresponding to the presence of $\beta$-sheet structure; however, due to noise, the positive peak at ~200 nm is not clear. Similarly, for NSP11-CoV2 aggregates (after 192 h incubation), CD spectrum reveals $\beta$-sheet rich structure showing the characteristic positive and negative ellipticity peaks at 198.5 and 218.5 nm, respectively. Spectral data for aggregates is smoothened with 7 points using the FFT filter function. For NSP11-CoV and NSP11-CoV2 monomers, data is smoothened with 7 and 6 points, respectively, using the same function. **c, h** Morphology of NSP11-CoV (45 days) and NSP11-CoV2

fibrils (192 h) is observed using tapping-mode AFM where the scale bar represents 700 and 100 nm, respectively. Five micrographs were captured from same sample for each peptide aggregate. **e** Aggregation kinetics of NSP11-CoV2, monitored using ThT fluorescence. Each dotted symbol represents the average value of three technical replicates and error bar represents the SEM of these replicates. The wine-coloured line shows the sigmoidal fitting of data points that determines the $T_{50}$ of this reaction to be $66.73 \pm 2.84$ h. **g** Under HR-TEM, the NSP11-CoV2 fibrils appear unbranched at 192 h; the scale bar represents 200 nm. Fifteen micrographs were captured from the same grid. **i** Height profile of an NSP11-CoV2 amyloid fibril shown with green-coloured line in panel (**h**). Additional micrographs and height profiles are given in Supplementary Fig. 4. The source data are given in the Source Data file. Arb. units are arbitrary units.

its homeostasis[55,57]. It also shortens the lag time of aggregation curves of major stress-granular proteins[58]. Furthermore, it can incorporate into the host stress granules impairing their self-disassembly in a manner related to amyotrophic lateral sclerosis[58].

Related to our work, some reports have demonstrated the ability of SARS-CoV-2 proteins to form amyloid aggregates. Small peptides derived from ORF6 and ORF10 proteins have been shown experimentally to form amyloid fibrils[59]. In addition, full-length ORF8 protein has been shown to form in vitro aggregates in cultured lung epithelial cells[60]. It is also speculated that APR-containing segments in viral

proteins can co-aggregate with cellular proteins and interfere with host pathways altering protein homeostasis in infected cells[5,8,61].

In view of these reports, to further increase our understanding of the possible amyloid nature of SARS proteins, we studied the complete proteomes of SARS-CoV and SARS-CoV-2. The focus of our study was to investigate the intrinsic propensity of all types of SARS proteins (structural, accessory, and non-structural) to form amyloid aggregates under near-physiological in vitro conditions. Our results prompt further investigations of the possible role of the aggregation of viral proteins in the range of pathologies induced by SARS-CoV-2.

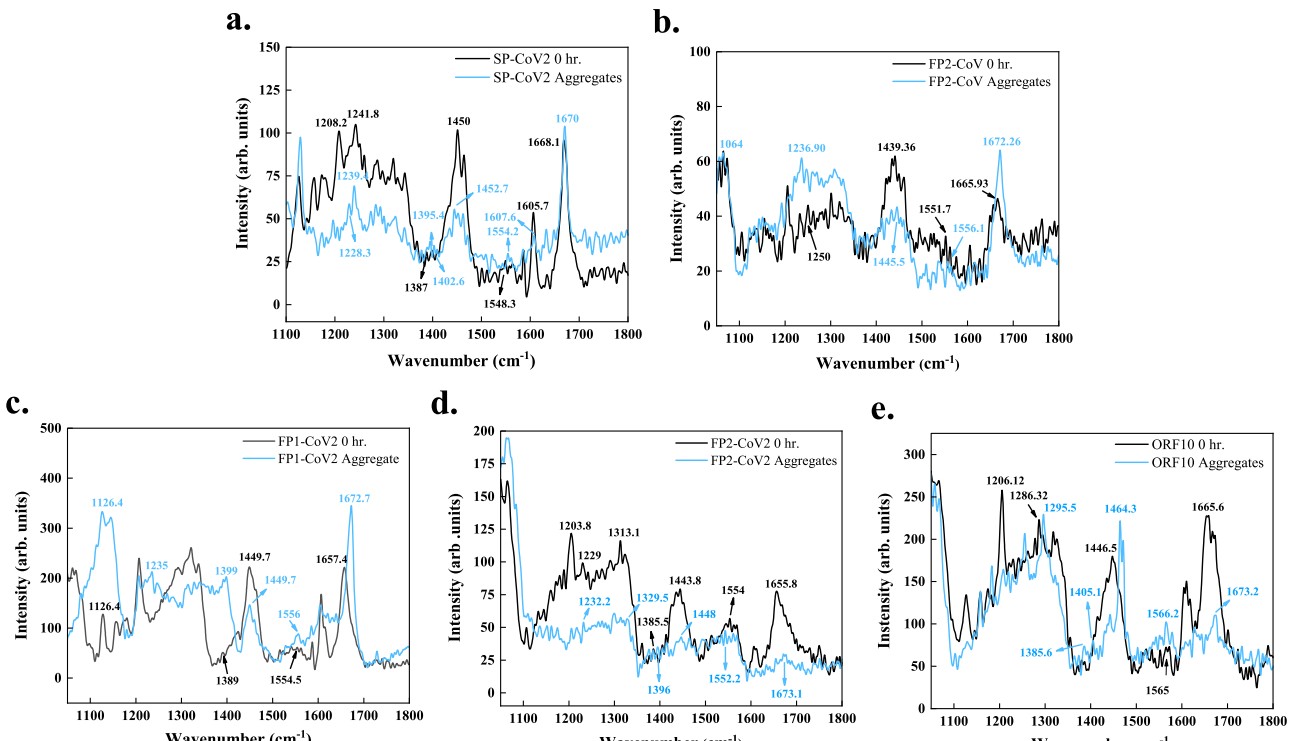

**Fig. 8 | Amyloid formation of SARS-CoV and SARS-CoV-2 peptides probed using Raman spectroscopy.** Raman spectra were acquired on excitation at 633 nm from wavenumber 1050 cm⁻¹ to 2000 cm⁻¹ for monomers and aggregates of **a** SP-CoV2, **b** FP2-CoV, **c** FP1-CoV2, **d** FP2-CoV2, and **e** ORF10. Apparent shifts in the positioning of various bands in spectra of aggregates from monomers were analyzed. Particularly, shifts in amide I-III bands and the Cα–H band are reported (see Supplementary Table 10). The spectral data was smoothened with 10 points using the FFT filter. The source data are given in the Source Data file. Arb. units are arbitrary units.

## Methods

### Peptides

The following proteins and peptides were chemically synthesized from GenScript, USA: SARS-CoV S fusion peptides (FP1-CoV, residues 798–819, 97.5% purity; FP2-CoV, residues 817–837, 90.7%), SARS-CoV NSP11 protein (NSP11-CoV, residues 4370–4382, 92.9% purity). SARS-CoV-2 S signal sequence (SP-CoV2, residues 1–12, 92.6% purity), SARS-CoV-2 S fusion peptides (FP1-CoV2, residues 816–837, 79% purity; FP2-CoV2, residues 835–855, crude), SARS-CoV-2 NSP6 residues 91–112 (NSP6-p, 88.5% purity), SARS-CoV-2 NSP11 protein (NSP11-CoV2, residues 4393–4405, 72.9% purity). SARS-CoV-2 ORF10 protein (full length, crude) was purchased from Thermo Scientific, USA. Mass spectrometry data of all the peptides are included in Supplementary Dataset and the information of synthesis is given in Supplementary Table 12.

**Chemicals.** TEM Grids and MICA sheets were obtained from Ted Pella Inc., USA. Other chemicals, including ammonium molybdate, thioflavin T (ThT), and 3-(4,5-dimethylthiazol-2-yl)−2,5-diphenyltetrazolium bromide (MTT) were procured from Sigma Aldrich, St. Louis, USA. The chemicals used in the cell culture study were purchased from Gibco™. The cell lines SH-SY5Y neuroblastoma (catalogue number: CRL-2266) and HepG2 (catalogue number: HB-8065) were obtained from the National Centre for Cell Sciences (NCCS), Pune, India.

**Prediction of amyloidogenic regions in SARS-CoV and SARS-CoV-2 proteome.** APRs were predicted by four sequence-based methods. MetAmyl, AGGRESCAN, FoldAmyloid, and FISH Amyloid. AGGRESCAN is based on a scale for natural amino acids derived from in vivo experiments. It also assumes that short and specific sequences within the protein can regulate protein aggregation. It gives a hot spot area (HSA) score for susceptible aggregate-forming residues[62].

FoldAmyloid calculates the backbone-backbone hydrogen bond formation probability and efficiently classifies the amyloidogenic peptides. It determines the amyloidogenic residues scoring above 21.4, a threshold assumed by server[63]. MetAmyl is a meta-predictor and combines the strength of different individual predictors—PAFIG, SALSA, Waltz, and FoldAmyloid. It creates a logistic regression model and gives a score interpreted as the probability of a fragment forming an amyloid fibril[64]. FISH Amyloid is a machine learning prediction method based on the presence of a segment with the highest scoring for co-occurrence of residue pairs[65]. Additionally, CamSol is used to predict proteins' hydrophilic and hydrophobic regions. It calculates the intrinsic solubility of proteins, which is inversely related to their aggregation propensity. CamSol assigns values to each amino acid; negative values below −1 represent insoluble residues, and positive values above +1 represent soluble residues[66]. The region in a protein that is predicted with five or more amino acids long is considered an amyloidogenic region in this study. In addition, the mean percentage of APRs calculated using predictors and average profile value from the FoldAmyloid server for SARS-CoV-2 proteins is also reported in Fig. 1c, d. Furthermore, in silico mapping of 20S proteasome cleavage sites across SARS-CoV-2 proteome is predicted with proteasome prediction web server NetChop 3.1[67].

**Preparation of the samples for the aggregation assays.** Peptides were dissolved in 100% 1,1,1,3,3,3-hexafluoro-2-propanol (HFIP) to remove pre-existing aggregates left to evaporate at room temperature overnight to get dry peptides. The peptide films were then dissolved according to their hydrophobic character and solvent recommended by GenScript and Thermo Scientific, USA, before incubation for aggregation experiments. S fusion peptides 1 and 2 of SARS-CoV were soluble in ultrapure water. All the other peptides were soluble in DMSO except NSP11 proteins that were soluble in buffer. After dissolving the

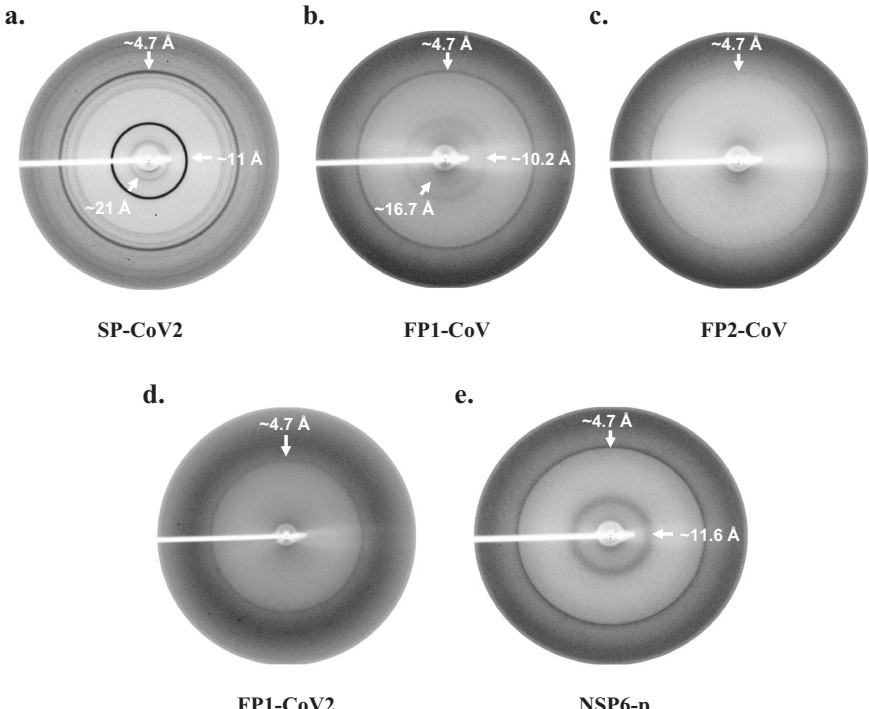

**Fig. 9 | X-ray diffraction pattern of un-oriented amyloid aggregates of SARS-CoV and SARS-CoV-2 peptides.** X-ray diffraction data for peptides **a** SP-CoV2, **b** FP1-CoV, **c** FP2-CoV, **d** FP1-CoV2, and **e** NSP6-p were collected at room temperature as described in the methods section. Prominent X-ray diffraction rings at ~4.7 Å and ~10–12 Å were observed in the samples. The sample-to-detector distance was kept at 350 mm for all the samples except FP1-CoV2, where this distance was kept at 300 mm during data collection.

monomeric peptides, samples for the aggregation assays were incubated at 37 °C with constant stirring (1000 rpm) on Eppendorf ThermoMixer C. Details of peptides and proteins used for in vitro aggregation assay are given in Supplementary Table 11.

**Thioflavin T aggregation assay.** To analyze the aggregation process in vitro, we used thioflavin T (ThT), an extensively used dye that, on binding to amyloid fibrils, shows an increase in fluorescence[68,69]. Totally, 25 μM samples were prepared in 20 mM sodium phosphate buffer (pH 7.4) with 25 μM ThT and incubated for 5 min in dark conditions for SP-CoV2, FP1-CoV2, FP2-CoV2, ORF10, NSP6-p, NSP11-CoV peptides. For FP1-CoV, 30 μM ThT and 10 μM peptide, for FP2-CoV, 20 μM ThT and 10 μM peptide, and for NSP11-CoV2, 20 μM ThT with 50 μM peptide was used. Water instead of buffer was used during the experiments for both SARS-CoV fusion peptides. The use of high protein concentrations in vitro studies of aggregation reactions is commonly used to establish the mechanism of aggregation on a timescale amenable to experimental testing in the laboratory[30]. In vivo, where the concentrations are typically lower, the aggregation reaction would take place on a longer timescale. In this study, the ThT fluorescence scans of peptides were monitored on 440 nm excitation wavelength, and emission spectra were recorded from 460 nm to 600 nm wavelengths using the black 96-well plates in TECAN Infinite M200 PRO multimode microplate reader. For NSP11-CoV2 peptide, ThT fluorescence scans were recorded on excitation at 450 nm wavelength using a Horiba Fluorolog-3 spectrofluorometer. For DMSO soluble peptides, ThT dye mixed with DMSO (at their respective concentration) and buffer was used as ThT control (for information of buffers, see Supplementary Table 11).

Kinetics experiments of all peptides were investigated using the ThT fluorescence excitation/emission at 440/490 nm using the black 96-well plates in TECAN Infinite M200 PRO multimode microplate reader. All the measurements were set up in duplicates or triplicates

and the average value was reported with standard error (SEM). Data were fitted using a sigmoidal curve to obtain the $T_{50}$ (the time at which ThT fluorescence intensity reaches 50% of its maximum value) value from the equation:

$$y = A2 + \frac{A1 - A2}{1 + e^{(x - x_0)/dx}} \tag{1}$$

Where A1 indicates the initial fluorescence, A2 the final fluorescence, $x_0$ the midpoint ($T_{50}$) value and $dx$ is a time constant. The lag phase of the reaction, denoted by $T_{lag}$, is calculated using the equation:

$$T_{lag} = T_{50} - 2(dx) \tag{2}$$

**CD spectroscopy.** The far-UV CD spectra were recorded in a 1 mm path length quartz cuvette at 25 °C from 190 nm to 240 nm wavelength range at every 0.5 nm. Biologic's MOS-500 spectrophotometer was used for acquiring the spectral data of all the peptides except NSP11-CoV and NSP11-CoV2 peptides for which JASCO J-1500 spectrophotometer system was employed. In both spectrophotometer, spectral data were acquired by two/three consecutive acquisitions and the averaged spectra is reported. CD spectra of monomeric peptides were measured in their respective solvents (see Supplementary Table 11 for solvents). For DMSO soluble peptides, spectral data of monomers were not obtained due to high HT voltage values. Spectra of aggregated peptides were obtained after removing DMSO by washing the aggregates twice in 20 mM sodium phosphate buffer (pH 7.4). For NSP11-CoV2 peptide, 20 mM sodium phosphate buffer (pH 7.4) containing 50 mM NaCl was used. The spectrum of the buffer was used to normalize the spectrum of respective samples.

**Atomic force microscopy (AFM).** The AFM images of aggregated fibrils were obtained using tapping-mode AFM (Dimension Icon from

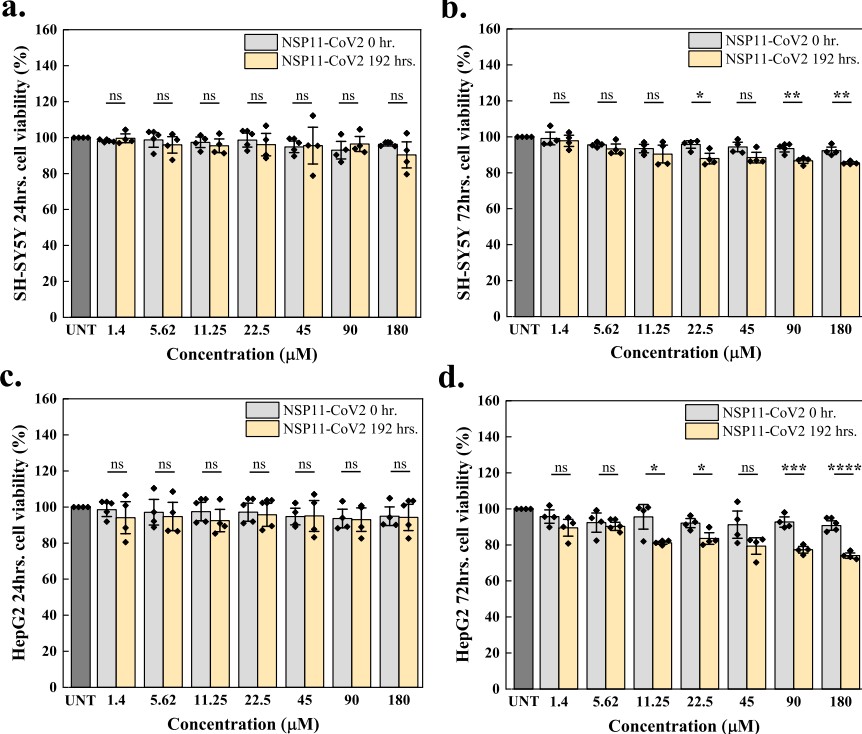

**Fig. 10 | Cell viability upon NSP11-CoV2 aggregation using MTT assay.** SH-SY5Y and HepG2 cells were treated with NSP11-CoV2 monomers (grey bars) and amyloid fibrils (192 h incubation; light yellow) for 24 h (**a**, **c**) and 72 h (**b**, **d**). The cells treated with media containing 20 mM sodium phosphate buffer (pH 7.4) were used as a control (dark grey bar; UNT is untreated). Diamond symbols on each bar indicate technical replicates and the error bars represent the SEM of these replicates. ns is a non-significant difference, *p < 0.05, **p < 0.01, ***p < 0.001, and ****p < 0.0001 according to the two-tailed Student's t-test (no adjustments were made).

Correspondingly, for 24 and 72 h treatment on SH-SY5Y cells, the respective p-values of monomer vs. aggregate are 0.8523, 0.7277 (1.4 μM), 0.5860, 0.3560 (5.62 μM), 0.6837, 0.4781 (11.25 μM), 0.4924, 0.0289 (22.5 μM), 0.9899, 0.0779 (45 μM), 0.5091, 0.0050 (90 μM), and 0.3101, 0.0063 (180 μM). For 24 and 72 h treatment on HepG2 cells, the respective p-values of monomer vs. aggregate are 0.1966, 0.0806 (1.4 μM), 0.5306, 0.6118 (5.62 μM), 0.0513, 0.0255 (11.25 μM), 0.1963, 0.0265 (22.5 μM), 0.9911, 0.0897 (45 μM), 0.8042, 0.0001 (90 μM), and 0.8208, <0.0001 (180 μM). The source data are given in the Source Data file.

Bruker). The measurements were carried out by depositing a twenty- to thirty-fold diluted solution of aggregated samples on a freshly cleaved mica surface. After incubation for 1 h, the surfaces were rinsed with deionized water, dried at room temperature overnight, and images were recorded.

**High resolution-transmission electron microscopy (HR-TEM).** The images were obtained using the in-house HR-TEM equipment FP 5022/22-Tecnai G2 20 S-TWIN, FEI. A 30-fold diluted sample of aggregates was drop-casted on copper grids coated with carbon (200-mesh; Ted Pella, Inc, USA) and stained negatively with 3% ammonium molybdate. The samples were then allowed to dry overnight before capturing the images.

**Raman spectroscopy.** Raman spectra of aggregates were captured using the home source Horiba's LabRAM HR Evolution Raman spectrometer equipped with different lasers from wavenumber 1050 cm⁻¹ to 2000 cm⁻¹. The excitation laser of 633 nm with a grating of 1800 grooves/mm was used. The power source was kept at 17 mW, and acquisitions were acquired for 30-60 sec for each measurement. Baseline measurements were performed using the in-built function in LabSpec 6 software. The spectra were calibrated using a standard silicon wafer before measurements. Monomeric peptides of FP2-CoV, FP1-CoV2, FP2-CoV2, and ORF10 were dissolved in HFIP and then drop-casted into a thin film on a silicon wafer to obtain the spectra. For SP-CoV2 and FP1-CoV monomers, powdered peptides were used to measure the Raman scattering. Aggregates of DMSO soluble peptides were washed twice with 20 mM sodium phosphate buffer (pH 7.4)

before being drop-casted on a silicon wafer. Aggregates of buffer soluble peptides were directly used.

**X-ray diffraction (XRD).** The preformed fibril samples were centrifuged at 5000g to the pellet and were washed and resuspended in 20 mM sodium phosphate buffer (pH 7.4). The samples were mixed by tapping and transferred to the 0.6 mm diameter borosilicate glass capillary (Hampton Research, UK) and sealed with clay from the narrow end. The samples were air-dried to remove excess solution to reduce noise in the XRD experiments. The data was collected using the in-house source Rigaku Micromax 007 equipped with a Mar345 detector at 1.5418 Å wavelength at room temperature. The air-dried samples were exposed for 25–30 min under the X-ray beam.

**Cell viability assays.** The effect of NSP11-CoV2 fibrils on cell viability was assessed on two cell lines (SH-SY5Y neuroblastoma cells and HepG2 hepatocellular carcinoma cells) using a colorimetric assay that uses a reduction of a yellow tetrazolium salt MTT to purple formazan crystals by metabolically active cells. SH-SY5Y cells were maintained in Dulbecco's modified Eagle's medium/ Nutrient Mixture F-12 Ham (DMEM F-12), while HepG2 cells were maintained in DMEM, both supplemented with 10% fetal bovine serum. Monomeric (0 h incubation) and aggregated (192 h incubation) samples of NSP11-CoV2 were resuspended in the calculated media volume to attain the desired concentration. Further, 6000 cells/well in a 96-well plate were seeded with 150 μl volume in culture media and incubated for 24 h at 37 °C for surface adherence. After incubation, cells were treated with varying

concentrations of NSP11-CoV2 (1.4–180 μM) samples in 100 μl fresh culture media and incubated at 37 °C for 24 h and 72 h. MTT (0.5 mg/ml final concentration) was added to each well and incubated for 3 h at 37 °C, and then 100 μl DMSO was added to dissolve the formazan crystal. The absorbance was recorded at 590 nm (630 nm reference wavelength) in TECAN Infinite M200 PRO multimode microplate reader. Data were collected in quadruplets and reported as the relative percentage of cell viability with respect to untreated cells.

**Statistical analysis and data representation.** All data were plotted and analysed using the OriginLab 8 software. For MTT assay data analysis, the student's *t*-test (two-tailed) was applied between the monomer and aggregate groups to compare significant cell survival using GraphPad Prism 8.0.2.

### Reporting summary

Further information on research design is available in the Nature Portfolio Reporting Summary linked to this article.

## Data availability

All data are contained within the manuscript or as supporting information. Source data files associated with all the graphs are also provided with this paper. The protein/peptide sequences are retrieved from NCBI (https://www.ncbi.nlm.nih.gov/) and UniProt (https://www.uniprot.org/) databases. All the accession codes are provided in the respective sections of proteins in supplementary tables 1 to 6. Source data are provided with this paper.

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

## Acknowledgements

Authors are thankful to the Indian Institute of Technology Mandi (BioX, AMRC, and C4DFED centres) for all the facilities and the faculty research grant, School of Biosciences and Bioengineering, IIT Mandi to R.G. T.B. is thankful to the Department of Science and Technology (DST) for the INSPIRE Fellowship. K.G. and P.K. are supported by the Science and Engineering Research Board (SERB), India (Grant Number: CRG/2019/005603). K.U.S. and S.K.K. are supported by the Indian Council of Medical Research (ICMR), India, for senior research fellowships. A.K., R.J., B.M., and A.B. are supported by the Ministry of Human Resource Development (MHRD). NG acknowledges the seed grant under the Institute of Eminence Scheme, Banaras Hindu University, for financial support. R.G. is grateful for the IYBA Award (Grant No. BT/11/IYBA/2018/06) from the Department of Biotechnology (DBT), India; MHRD-SPARC (SPARC/2018-2019/P37/SL), SERB, India (Grant No. CRG/2019/005603), and ICMR (Grant Nos. 58/6/2020/PHA/BMS and 52/04/2020/BIO/BMS).

## Author contributions

R.G., N.G., and M.V.: Conception, design, review, data analysis, and paper writing. T.B., K.G., P.K., N.N., S.K.K., R.J., B.M., A.K., Z.F.B., A.B., and K.U.S. performed experiments and computational analysis predictions. T.B., K.G., S.K.K., and P.K. analyzed data and wrote the paper. K.G.T. obtained the XRD patterns in the study.

## Competing interests

The authors declare no competing interests.
