## [Peer Review File · Nature Communications]

Amyloidogenic proteins in the SARS-CoV and SARS-CoV-2 proteomesREVIEWER COMMENTS

Reviewer #1 (Remarks to the Author):

This study comprehensively searched amyloidogenic regions in SARS-CoV-2, comparing with SARS-CoV. Additionally, this research characterized the amyloid-like propensity of those peptides, using thioflavin assay, atomic force microscopy, and circular dichroism. Furthermore, cell viability in the presence of one of those peptides, NSP11, toxicity was assessed at the cell level.

Main

1. This study searched for possible proteasome cleavage sites, but have fragmented peptides actually been identified in aggregates in cultured cells or in vivo? Clarifying this question will help the reader better understand the relationship between this study and pathology.
2. To conclude the amyloid-like propensity of the novel peptide regions identified, the seeding activity of their fibrils should be demonstrated. Does the addition of (fragmented) fibrils reduce the lag times on the ThT time courses?
3. Related to both questions 1 and 2, does a partial peptide co-aggregate with a longer (or full-length) peptide containing itself? For example, the partial peptides of the tumor suppressor, p53, co-aggregate with the longer p53 DNA-binding region (Ghosh et al. *Biochemistry*. 2014, doi: 10.1021/bi500825d). The partial peptides of type 2 diabetes-causing amyloidogenic peptide, amylin, can be co-aggregated with full-length peptides (Kakinen et al. *Nano Lett.* 2019, doi: 10.1021/acs.nanolett.9b02771).
4. Do the circular dichroism spectra after aggregation of peptides other than those in Figures 5 and 10 show a β -sheet structure? As with other amyloids, AFM images appear to include structures other than fibrils (e.g., oligomers). ThT shows time variation in the proportion of cross β structures in the aggregation reaction system, but does not necessarily guarantee that the aggregating structure is fibrous. It is necessary to show either that β -sheet structures are dominant throughout the system in the CD spectrum or more directly whether fibrils have amyloid-specific structures (e.g., labeling with an amyloid structure-specific antibody (OC)).

Minor

1. The correspondence of the closed and open boxes in Figure 1b is unclear due to the horizontal gaps between the open and closed boxes. It is better to delete the open boxes corresponding to "S" to "10" and put them in the closed boxes.
2. Figure 2a and b would be easier to understand the difference between the two if the values corresponding to both were displayed on one graph with markers, rather than separately in a bar chart.
3. The fibrils shown in Figure 4d appear to be doubled in the vertical direction. This should be a typical double-tip effect specific to AFM imaging. This should be described in the figure caption to make it easier for the readers to understand.
4. From the comprehensively searched peptide sequences, the reasons for focusing on spike signal peptides and spike protein fusion peptides 1 and 2, ORF10 protein, NSP6-p, and NSP11 protein as targets for in vitro experiments and NSP11 as a MTT assay target should be described.

Reviewer #2 (Remarks to the Author):

The present study outlines the presence of aggregation-prone regions in many SARS-CoV-2 proteins. The investigators have chemically synthesized these regions and have performed in vitro studies to confirm the prediction. To some extent, the investigators have successfully shown that the short sequences formed amyloid-like structures. However, the pathological relevance of this study is very much limited, due to incomplete information about the existence of these peptides during the disease manifestation. A number of recent publications have already identified the aggregation prone regions in almost every SARS-CoV-2 protein. For more information, authors may refer to the following publications; *Biochim Biophys Acta Proteins Proteom.* 2021 Oct; 1869(10):140693.

doi:10.1016/j.bbapap.2021.140693. Epub 2021 Jul 5; J Proteins Proteom 12, 1–13 (2021).
<https://doi.org/10.1007/s42485-021-00057-y>).

This could be one dimension of SARS-CoV-2 Proteome. However, so far, no study has confirmed the amyloidogenic character of the viral proteome play any role in defining the COVID-19 pathogenesis.

The enhanced probability of amyloid formation by host proteins (for example, serum amyloid A and other CRP proteins) followed by SARS-CoV-2 infection is indicated as a secondary effect. Hence, until and unless the presence of amyloids or amyloid-like aggregates derived from SARS-CoV-2 proteins is confirmed *in vivo*, the present study has limited relevance and values.

On the other hand, the mere presence of aggregation-prone regions in proteins does not render them amyloidogenic and pathological.

Authors are suggested to undertake the following recommendations and perform additional experiments to improve the relevance and overall quality of the manuscript;

Major

1. There is no evidence that the viral proteins/ peptides form amyloid *in vivo*. Authors must confirm the existence of these amyloids in *in vivo* system.

2. Authors must use an animal cell model system for SARS-CoV-2 infection and try to confirm the presence of amyloid formation. Subsequently, they should examine the same in relevant animal models to make the finding definitive and affirmative.

3. Nothing is mentioned about the solubility and critical concentrations of all the proteins/ peptides. Whether the concentration/s used for aggregation is/ are practically possible under *in vivo* conditions? Authors are required to clarify that.

4. Authors have used only Thioflavin T binding assay to characterize the amyloid nature of protein aggregates. It cannot be taken for granted to confirm the amyloid character of the protein aggregates. Authors are suggested to apply X-ray diffraction and amyloid-specific antibody staining.

5. There is a lack of a mechanistic model for cytotoxicity. Whether the observed cytotoxicity is due to oligomers or fibrils? Authors are suggested to use appropriate methods to confirm the nature of amyloid conformers. Based upon that, they should build a mechanistic model.

Minor

1. The AFM image in figures 7 and 8 look like an artifact. Instead, TEM imaging can be performed.

Reviewer #3 (Remarks to the Author):

In the present study, authors predicted the amyloidogenic proteins in the SARS-CoV and SARS-CoV-2 proteomes. The results of the present study depict the aggregation tendency of several proteins in the SARS-CoV and SARS-CoV-2 proteomes. The authors need to address the following queries:

1. Page 5, "According to our analysis, AGGRESCAN detected a total of 18% amyloidogenic regions in SARS-CoV-2 N and only 16% in SARS-CoV N." The authors need to mention what does N indicate in SARS-CoV-2 N and SARS-CoV N?

2. On page 8, authors mentioned, "Also, fusion peptides of SARS-CoV-2 have highly aggressive aggregation reactions than their SARS-CoV counterparts." The authors need to discuss the possible reasons why fusion peptides of SARS-CoV-2 display higher aggregation tendency as compared to fusion peptides of SARS-CoV?

3. For the better clarity of the readers, authors need to mention the concentration of the FP2-CoV and NSP11 used in the CD spectroscopic studies on page 7 and 9, respectively.

4. Page 25, "Figure 5. *In vitro* aggregation of fusion peptide 2 (FP2-CoV) of the SARS-CoV spike

protein." It is not entirely clear why FP2-CoV aggregates were observed in 104 hours using ThT assay, whereas it took 720 hour in case of CD spectra? The authors need to mention changes in the secondary structure of FP2-CoV at 104 hours.

5. The authors need to mention the basis for the selection of NSP11 aggregates for the cytotoxicity studies on the mammalian cell lines. Why FP1-CoV2 having a T50 of about 1 hour compared to T50 of 64 hours for NSP11 was not selected for the cytotoxicity studies on the mammalian cell lines?

6. Page 10, correct "facilitatation" as "facilitation".

7. The authors need to comment on the purity of the synthesized peptides used in the ThT and CD studies. The HPLC profiles of the synthesized peptides need to be incorporated in the supporting information.

8. The authors need to provide the page numbers in references 1, 6, 7, 55, 56 etc. The journal name needs to be in the same format e.g. reference 44 and 47, journal name is in capital letters.

RESPONSE TO REVIEWERS

Reviewer 1:

This study comprehensively searched amyloidogenic regions in SARS-CoV-2, comparing with SARS-CoV. Additionally, this research characterized the amyloid-like propensity of those peptides, using thioflavin assay, atomic force microscopy, and circular dichroism. Furthermore, cell viability in the presence of one of those peptides, NSP11, toxicity was assessed at the cell level.

Major comments:

1. This study searched for possible proteasome cleavage sites, but have fragmented peptides actually been identified in aggregates in cultured cells or in vivo? Clarifying this question will help the reader better understand the relationship between this study and pathology.

Authors' response: We are thankful to the reviewer for asking this question. We searched the literature regarding the aggregation of SARS coronavirus proteins, finding a recent paper (Ref 67 in the revised version of our manuscript) where the full-length ORF8 protein of SARS-CoV-2 was shown to form aggregates in cultured lung epithelial cells. In our study, we analyzed this protein using MetAmyl, which is a meta-predictor and provides APRs based on combining the strength of four different algorithms used by PAFIG, SALSA, Waltz, and FoldAmyloid. Our results show that this contains multiple APRs. These identified residues possibly be driving the aggregation of ORF8 in cultured cells as studied in Ref 67.

2. To conclude the amyloid-like propensity of the novel peptide regions identified, the seeding activity of their fibrils should be demonstrated. Does the addition of (fragmented) fibrils reduce the lag times on the ThT time courses?

Authors' response: We thank the reviewer for this suggestion. We performed experiments demonstrating the seeding activity of preformed fibrils (PFF) with proteins associated with amyloidogenic diseases. As shown in figure below, the effects of FP1-CoV2 and FP2-CoV2 seeds on aggregation of amylin demonstrate their seeding activity. According to the results, FP2-CoV2 seeds have accelerated the aggregation kinetics of amylin. It is affecting the T_{50} as well as T_{lag} of amylin aggregation reaction in vitro. These results may provide a possible link between COVID-19 with type II diabetes. However, we have not included these results in the revised manuscript since more systematic investigations should be carried out to establish firmly this connection.

Reactions	T ₅₀ (min)	T _{lag} (min)
Amylin control (5 μM)	21.04	15.12
Amylin (5 μM) + FP1-CoV2 seeds (1 μM)	20.87	9.67
Amylin (5 μM) + FP2-CoV2 seeds (1 μM)	13.93	11.01

Figure: Effects of FP1-CoV2 and FP2-CoV2 seeds on an aggregation of Amylin.

3. Related to both questions 1 and 2, does a partial peptide co-aggregate with a longer (or full-length) peptide containing itself? For example, the partial peptides of the tumor suppressor, p53, co-aggregate with the longer p53 DNA-binding region (Ghosh et al. Biochemistry. 2014, doi: 10.1021/bi500825d). The partial peptides of type 2 diabetes-causing amyloidogenic peptide, amylin, can be co-aggregated with full-length peptides (Kakinen et al. Nano Lett. 2019, doi: 10.1021/acs.nanolett.9b02771).

Authors' response: We thank reviewer for this interesting suggestion. However, we could not add this experiment to this study due to the unavailability of the full-length proteins.

4. Do the circular dichroism spectra after aggregation of peptides other than those in Figures 5 and 10 show a β -sheet structure? As with other amyloids, AFM images appear to include structures other than fibrils (e.g., oligomers). ThT shows time variation in the proportion of cross β structures in the aggregation reaction system but does not necessarily guarantee that the aggregating structure is fibrous. It is necessary to show either that β -sheet structures are dominant throughout the system in the CD spectrum or more directly whether fibrils have amyloid-specific structures (e.g., labeling with an amyloid structure-specific antibody (OC)).

Authors' response: We are grateful to the reviewer for this suggestion. We have performed CD experiments with most of the peptide aggregates (however, due to lack of solubility we could not use CD on the monomeric peptides of SP-CoV2, FP1-CoV2, FP2-CoV2, and ORF10 (Figures 3-8, 10 and 11)).

We have also performed Raman spectroscopy on the monomeric peptides and aggregates of most of the peptides (Figures 12 and Table 1). Since Raman spectra are considered a molecular fingerprint, the shift in these spectra between the monomeric and aggregated proteins is widely used to characterize the formation of amyloid aggregates. (Ramachandran et al. Biochemistry 53, 6550, 2014; Flynn et al. J. Biol. Chem. 293, 776, 2018).

Minor comments:

1. The correspondence of the closed and open boxes in Figure 1b is unclear due to the horizontal gaps between the open and closed boxes. It is better to delete the open boxes corresponding to "S" to "10" and put them in the closed boxes.

*Author's response: We have followed this suggestion and modified **Figure 1b** in the revised manuscript.*

2. Figure 2a and b would be easier to understand the difference between the two if the values corresponding to both were displayed on one graph with markers, rather than separately in a bar chart.

*Author's response: We have revised **figure 2** and made separate graphs comparing the SARS-CoV and SARS-CoV-2 (a) structural, (b) accessory, and (c) non-structural proteins. Although, there are some differences in proteins such as ORF8 protein of CoV2 is present in two parts ORF8a and ORF8b in CoV1. and ORF10 is not present in CoV1. The figure legend has also been modified.*

3. The fibrils shown in Figure 4d appear to be doubled in the vertical direction. This should be a typical double-tip effect specific to AFM imaging. This should be described in the figure caption to make it easier for the readers to understand.

*Author's response: We have mentioned the double-tip effect during AFM scanning which has caused the double images of fibrils in the figure legend of **figure 4d**.*

4. From the comprehensively searched peptide sequences, the reasons for focusing on spike signal peptides and spike protein fusion peptides 1 and 2, ORF10 protein, NSP6-p, and NSP11 protein as targets for in vitro experiments, and NSP11 as a MTT assay target should be described.

Authors' response: The SARS-CoV and SARS-CoV-2 proteins have multiple APRs which can derive the aggregation of full-length proteins. Here, we considered peptides from each type of protein i.e. at least one from structural, non-structural, and accessory proteins for further experimental validation. Most of the peptides were DMSO soluble, therefore, we performed an MTT assay with NSP11 protein of SARS-CoV-2 as it can solubilize in sodium phosphate buffer.

Reviewer 2:

The present study outlines the presence of aggregation-prone regions in many SARS-CoV-2 proteins. The investigators have chemically synthesized these regions and have performed in vitro studies to confirm the prediction. To some extent, the investigators have successfully shown that the short sequences formed amyloid-like structures.

However, the pathological relevance of this study is very much limited, due to incomplete information about the existence of these peptides during the disease manifestation.

A number of recent publications have already identified the aggregation prone regions in almost every SARS-CoV-2 protein. For more information, authors may refer to the following publications; Biochim Biophys Acta Proteins Proteom. 2021 Oct; 1869(10):140693.

doi:10.1016/j.bbapap.2021.140693. Epub 2021 Jul 5; J Proteins Proteom 12, 1–13 (2021). <https://doi.org/10.1007/s42485-021-00057-y>.

This could be one dimension of SARS-CoV-2 Proteome. However, so far, no study has confirmed the amyloidogenic character of the viral proteome play any role in defining the COVID-19 pathogenesis.

The enhanced probability of amyloid formation by host proteins (for example, serum amyloid A and other CRP proteins) followed by SARS-CoV-2 infection is indicated as a secondary effect. Hence, until and unless the presence of amyloids or amyloid-like aggregates derived from SARS-CoV-2 proteins is confirmed in vivo, the present study has limited relevance and values.

On the other hand, the mere presence of aggregation-prone regions in proteins does not render them amyloidogenic and pathological.

Authors are suggested to undertake the following recommendations and perform additional experiments to improve the relevance and overall quality of the manuscript;

Major comments:

1. There is no evidence that the viral proteins/ peptides form amyloid in vivo. Authors must confirm the existence of these amyloids in in vivo system.

Authors' response: We agree with the reviewer that this is an important point, which we have now discussed in more detail in the revised version of the manuscript. We have described in particular a recent study (Geng et al., Frontiers in Immunology, 2021, now Ref 67 in the manuscript) that shows that the ORF8 protein of SARS-CoV-2 forms aggregates in cultured epithelial cells.

More generally, our study was inspired by the recent debate about the links between the presence of viruses and the onset of amyloid diseases (references 41-47, 57-60). Given the widespread nature of the COVID-19 pandemic, we believe that it is crucial to investigate these links in the case of SARS-CoV-2. With this in mind, we have also referred to other studies that have shown that the interaction of SARS-CoV-2 proteins with α -synuclein and A β 42 can enhance the aggregation of these proteins (references 61-64),

2. Authors must use an animal cell model system for SARS-CoV-2 infection and try to confirm the presence of amyloid formation. Subsequently, they should examine the same in relevant animal models to make the finding definitive and affirmative.

Authors' response: Reference 67 mentioned above reports the aggregation of a SARS-CoV-2 protein in a mammalian cell system. Although we could not within the scope of this study generalise this result to an animal model, we have revised the text to explain that our results, together with existing evidence, strongly suggest the importance of further investigations of the role of protein aggregation in the range of pathologies induced by SARS-CoV-2.

3. Nothing is mentioned about the solubility and critical concentrations of all the proteins/peptides. Whether the concentration/s used for aggregation is/ are practically possible under in vivo conditions? Authors are required to clarify that.

Authors' response: As suggested, we have added the information about the solubility and concentration of the peptides in revised manuscript in the sub-section 'Preparation of the samples for the aggregation assays'.

4. Authors have used only Thioflavin T binding assay to characterize the amyloid nature of protein aggregates. It cannot be taken for granted to confirm the amyloid character of the protein aggregates. Authors are suggested to apply X-ray diffraction and amyloid-specific antibody staining.

Authors' response: We are thankful to the reviewer for this suggestion. We have now performed XRD to observe the formation of β -sheet-rich amyloid fibrils of various peptide aggregates investigated in this study. We have added the detailed analysis of obtained XRD patterns in the revised manuscript in the section "X-ray diffraction pattern of aggregates"

5. There is a lack of a mechanistic model for cytotoxicity. Whether the observed cytotoxicity is due to oligomers or fibrils? Authors are suggested to use appropriate methods to confirm the nature of amyloid conformers. Based on that, they should build a mechanistic model.

Authors' response: We share the view of the reviewer that it would be important to understand whether the origin of the toxicity of the aggregates. As previously mentioned, we performed toxicity experiment with NSP11-CoV2 due to its solubility in sodium phosphate buffer. Although we could not perform the same experiments, with pure oligomers, due to the unavailability of peptides, our results show the toxicity of large aggregates.

Minor comments:

1. The AFM image in figures 7 and 8 look like an artifact. Instead, TEM imaging can be performed.

*Authors' response: We are grateful to the reviewer for pointing this out. We have followed this suggestion and performed TEM imaging of all the peptide aggregates investigated in this study. We have added the TEM images and modified the **figures 3-10** in the revised manuscript.*

Reviewer 3:

In the present study, the authors predicted the amyloidogenic proteins in the SARS-CoV and SARS-CoV-2 proteomes. The results of the present study depict the aggregation tendency of several proteins in the SARS-CoV and SARS-CoV-2 proteomes. The authors need to address the following queries:

1. Page 5, "According to our analysis, AGGRESCAN detected a total of 18% amyloidogenic regions in SARS-CoV-2 N and only 16% in SARS-CoV N." The authors need to mention what does N indicate in SARS-CoV-2 N and SARS-CoV N?

Author's response: As suggested, we have added the full forms for all the abbreviations used in the first place. In the mentioned sentences, 'N' refers to the Nucleocapsid proteins of SARS-CoV and SARS-CoV-2 viruses.

2. On page 8, the authors mentioned, "Also, fusion peptides of SARS-CoV-2 have highly aggressive aggregation reactions than their SARS-CoV counterparts." The authors need to discuss the possible reasons why fusion peptides of SARS-CoV-2 display higher aggregation tendency as compared to fusion peptides of SARS-CoV?

Author's response: We have explained the aggressive aggregation of SARS-CoV-2 spike fusion peptides over SARS-CoV based on the substitutions that occurred in SARS-CoV-2 spike protein. This has been added in the revised manuscript.

3. For the better clarity of the readers, authors need to mention the concentration of the FP2-CoV and NSP11 used in the CD spectroscopic studies on page 7 and 9, respectively.

*Author's response: As suggested, we have mentioned the concentrations of peptides (monomers and aggregates) used for the CD-spectroscopy. As CD spectra of other peptides are also added in the manuscript, we have written the concentration of the peptides used in their respective figure legends (**Figures 3-8, 10 and 11** in main manuscript).*

4. Page 25, "Figure 5. In vitro aggregation of fusion peptide 2 (FP2-CoV) of the SARS-CoV spike protein." It is not entirely clear why FP2-CoV aggregates were observed in 104 hours

using ThT assay, whereas it took 720 hour in case of CD spectra? The authors need to mention changes in the secondary structure of FP2-CoV at 104 hours.

*Author's response: We have now replaced the CD data of the FP2-CoV peptide. We have not tried obtaining the CD spectra at 104 hrs and we measured the CD data at 240 hrs while preparing the sample for XRD and Raman spectroscopy. Therefore, CD spectrum of 240 hrs aggregated FP2-CoV sample is replaced with the previous data at 720 hrs (see **Figure 5** in the revised manuscript).*

5. The authors need to mention the basis for the selection of NSP11 aggregates for the cytotoxicity studies on mammalian cell lines. Why FP1-CoV2 having a T50 of about 1 hour compared to T50 of 64 hours for NSP11 was not selected for the cytotoxicity studies on the mammalian cell lines?

Authors' response: As mentioned above, there was no specific reason to selectively choose the NSP11-CoV2 peptide of the MTT assay other than it is soluble in sodium phosphate buffer.

6. Page 10, correct "facilitatation" as "facilitation".

Author's response: We have corrected this typo.

7. The authors need to comment on the purity of the synthesized peptides used in the ThT and CD studies. The HPLC profiles of the synthesized peptides need to be incorporated in the supporting information.

Author's response: We have added the additional information and HPLC profiles of all synthesized peptides in the supplementary file in the section "High-performance liquid chromatography and mass spectrometry profiles of purified peptides"

8. The authors need to provide the page numbers in references 1, 6, 7, 55, 56 etc. The journal name needs to be in the same format e.g. reference 44 and 47, journal name is in capital letters.

Authors' response: We have made all the references in same format. Only ref 16 does not have page numbers.

REVIEWER COMMENTS

Reviewer #1 (Remarks to the Author):

The revised manuscript provides answers to the Reviewer's concerns to the extent possible, but the following points need to be clarified.

1. for Figures 3b, 6b, and 8b, where the plots are denser immediately after the start, an inset showing only the early reaction time period should be added to make it easier to see if there is a lag in the ThT time course.

2. in relation to 1, the presence or absence of seeding activity should be clearly indicated to give the reader an understanding of the amyloid-like properties of each protein. It is necessary to indicate whether the aggregates grow by incorporating the soluble structures of the proteins that make up each aggregate, rather than amylin. The change in lag time to elevated ThT fluorescence due to heterogeneous mixtures is not necessarily seeding activity (even metal ions change lag time). XRD, CD, and Raman analysis reveal amyloid structure, but not function. In Reviewer's experience, the protein aggregation reactions that increase ThT, the aggregation product may not necessarily decrease lag time. In the ThT time courses in Figure 3-10, it is sufficient to show whether mixing a plateau-phase sample with monomer solution decreases the lag time. This experiment can be performed with only the samples used in this study.

3. in the MTT assay, a significance criterion with an appropriate statistical test method should be provided.

Reviewer #2 (Remarks to the Author):

The authors have revised the manuscript, and its quality has improved somewhat. However, the bottleneck of the issue still remains unaddressed. For example, whether aggregation propensity and ability of the SARS-CoV-2 proteins/peptides are linked with its pathological outcomes. Or can the amyloids of SARS-CoV-2 proteins/peptides aggravate the neurodegeneration and cause Alzheimer's disease in the patients? No such experimental data in the manuscript can, even partially, support the hypothesis (this is what the authors have argued). Therefore, performing the seeding experiments, at least under in vitro conditions, and detecting the amyloid in biological samples derived from SARS-CoV-2 infected patients is indispensable.

Without that, the study will have limited relevance. It will not add any value to the existing knowledge, as many investigators have already confirmed the amyloidogenic potential and ability of SARS-CoV-2 proteins/peptides to form amyloids. The manuscript requires significant revision.

As mentioned before, a deliberation and input from clinicians handling COVID-19 might be of great help. The nasal secretions from the COVID-19 patients might constitute a viable biological material where the presence of amyloids could be looked upon. Investigators must realize that the bottleneck of this study is to establish the presence of the SARS-CoV-2 proteins/peptide-derived amyloid in vivo. Otherwise, merely reporting the amyloid-forming ability of the ORF6 and ORF10 transcripts might not be of great importance.

The other vital point here is to use the conformation-specific antibodies to substantiate the amyloid-forming ability of the protein/peptide aggregates derived from ORF6 and ORF10 transcripts.

The investigators are requested to address the points mentioned earlier by providing concrete experimental evidences rather than a generic approximation.

Reviewer #3 (Remarks to the Author):

In the revised manuscript, the authors have addressed the queries raised in the previous draft of the manuscript. In this work, the authors determined the amyloidogenic proteins in the SARS-CoV and SARS-CoV-2 proteomes.

1. On page 8 of the revised manuscript, the authors mentioned "A single negative peak near 218 nm provided evidence for the β -sheet rich structures in amyloids formed by SP-CoV2." A positive peak with a high ellipticity value at ~ 202 nm was observed in the CD spectra of SP-CoV2 aggregates (Fig. 3c), which was not observed in the aggregates formed by other proteins in the SARS-CoV-2 proteome. For better clarity of the readers, the authors need to discuss this in the revised manuscript.
2. In the section "X-ray diffraction pattern (XRD) of aggregates" on page 13 of the revised manuscript, the authors need to comment on whether any XRD pattern was observed in the case of ORF10 protein aggregates.
3. Page 14: "The aggregates formed by NSP6-p exhibit a classic meridional XRD reflection at ~ 4.7 Å." The authors need to mention why CD spectra were not recorded for the NSP6-p aggregates.

RESPONSE TO REVIEWERS

Reviewer 1:

The revised manuscript provides answers to the Reviewer's concerns to the extent possible, but the following points need to be clarified.

1. For Figures 3b, 6b, and 8b, where the plots are denser immediately after the start, an inset showing only the early reaction time period should be added to make it easier to see if there is a lag in the ThT time course.

Authors' Response: We thank the reviewer for this suggestion. As the kinetics of aggregation of some of the peptides are very fast, we have modified the kinetics graphs and added an inset in Figures 3b, 6b, and 8b showing the early time course of aggregation.

2. In relation to 1, the presence or absence of seeding activity should be clearly indicated to give the reader an understanding of the amyloid-like properties of each protein. It is necessary to indicate whether the aggregates grow by incorporating the soluble structures of the proteins that make up each aggregate, rather than amylin. The change in lag time to elevated ThT fluorescence due to heterogeneous mixtures is not necessarily seeding activity (even metal ions change lag time). XRD, CD, and Raman analysis reveal amyloid structure, but not function. In Reviewer's experience, the protein aggregation reactions that increase ThT, the aggregation product may not necessarily decrease lag time. In the ThT time courses in Figure 3-10, it is sufficient to show whether mixing a plateau-phase sample with monomer solution decreases the lag time. This experiment can be performed with only the samples used in this study.

Authors' Response: We are thankful to the reviewer for this suggestion. We have performed a self-seeding experiment with the FP2-CoV1 peptide at 0.5 mg/ml concentration. As seen from the image below (Figure 1), the 20% seeds added to the 0.5 mg/ml FP2-CoV1 have decreased their T_{50} value from 60.2 h (control) to 24.5 h. The results show the presence of seeding activity of the plateau-phase sample. We have also included these results in the revised supplementary information as Figure S1.

Figure S1: Seeding activity of FP2-CoV1 aggregates. As reported in the main manuscript, at 1 mg/ml concentration, T_{50} is 47.6 h. For 0.5 mg/ml concentration, T_{50} is 60.3 h. On addition of 20% FP2-CoV1 seeds, T_{50} decreases to 24.5 h.

3. In the MTT assay, a significance criterion with an appropriate statistical test method should be provided.

Authors' Response: We thank the reviewer for this comment. As suggested, we have performed a Student's t-test to analyse the MTT assay data. We have compared the significance between the monomeric control group and the aggregates on cell survival. We have also modified Figure 14, its legend, and its methodology in the revised manuscript.

Reviewer 2:

The authors have revised the manuscript, and its quality has improved somewhat. However, the bottleneck of the issue still remains unaddressed. For example, whether aggregation propensity and ability of the SARS-CoV-2 proteins/peptides are linked with its pathological outcomes. Or can the amyloids of SARS-CoV-2 proteins/peptides aggravate the neurodegeneration and cause Alzheimer's disease in the patients? No such experimental data in the manuscript can, even partially, support the hypothesis (this is what the authors have argued). Therefore, performing the seeding experiments, at least under in vitro conditions, and detecting the amyloid in biological samples derived from SARS-CoV-2 infected patients is indispensable. Without that, the study will have limited relevance. It will not add any value to the existing knowledge, as many investigators have already confirmed the amyloidogenic potential and ability of SARS-CoV-2 proteins/peptides to form amyloids. The manuscript requires significant revision. As mentioned before, a deliberation and input from clinicians handling COVID-19 might be of great help. The nasal secretions from the COVID-19 patients might constitute a viable biological material where the presence of amyloids could be looked upon. Investigators must realize that the bottleneck of this study is to establish the presence of the SARS-CoV-2 proteins/peptide-derived amyloid in vivo. Otherwise, merely reporting the amyloid-forming ability of the ORF6 and ORF10 transcripts might not be of great importance. The other vital point here is to use the conformation-specific antibodies to substantiate the amyloid-forming ability of the protein/peptide aggregates derived from ORF6 and ORF10 transcripts. The investigators are requested to address the points mentioned earlier by providing concrete experimental evidences rather than a generic approximation.

Authors' Response: We are in agreement with the reviewer for his/her concern over detecting the viral protein amyloids in vivo. However, we would like to clarify that the focus of our study is to investigate the potential of amyloid formation by SARS coronavirus proteins and peptides, which is also clear from the title of the manuscript. We have not focused on the pathological aspect of SARS coronavirus causing neurodegenerative diseases. The question, however, remains open for future studies, which we hope will be stimulated by our work.

We agree with the reviewer that we do not demonstrate that SARS-CoV or SARS-CoV-2 protein amyloids cause Alzheimer's Disease (AD) in infected patients. Therefore, we have added a

paragraph in the Discussion section detailing the limitations of this study. Also, we removed that previous Figure S1, which suggested a sequence similarity between the spike protein of SARS-CoV-2 and A β 42. In addition, other paragraphs of the discussion are extensively modified according to the perspective of peptide aggregation. Therefore, all the speculations suggestive of SARS-CoV and CoV2 protein aggregation causing neurological complications are either modified or removed.

We would also like to clarify that our manuscript is a systematic in-vitro study on the aggregation of SARS-CoV and SARS-CoV-2 proteins and peptides. We have used more than five different methods to show and prove the conformation of the amyloid formation, such as dye-based assay, imaging techniques like TEM, and AFM, spectroscopic techniques like CD and Raman spectroscopy, and also X-ray diffraction. Therefore, we may not need to use another method of conformation-specific antibodies to demonstrate the amyloid-forming ability.

Reviewer 3:

In the revised manuscript, the authors have addressed the queries raised in the previous draft of the manuscript. In this work, the authors determined the amyloidogenic proteins in the SARS-CoV and SARS-CoV-2 proteomes.

1. On page 8 of the revised manuscript, the authors mentioned “A single negative peak near 218 nm provided evidence for the β -sheet rich structures in amyloids formed by SP-CoV2.” A positive peak with a high ellipticity value at \sim 202 nm was observed in the CD spectra of SP-CoV2 aggregates (Fig. 3c), which was not observed in the aggregates formed by other proteins in the SARS-CoV-2 proteome. For better clarity of the readers, the authors need to discuss this in the revised manuscript.

Authors' Response: *We thank the reviewer for this comment. The positive peak at \sim 202 nm in CD spectra of SP-CoV2 also corresponds to the β -sheet structure. Similar positive peaks are also present in spectral data of FP1-CoV, ORF10, and NSP11-CoV2 aggregates at 204.5 nm, 203.5 nm, and 198.5 nm, respectively. In the case of FP2-CoV, due to high negative signal at 216 nm, the positive peak at 201.5 nm has overall negative ellipticity. For FP1-CoV2, the positive signal near 200 nm is unclear due to the noise in data. For FP2-CoV2, due to low signal, the peak near 207 nm shows overall negative ellipticity. However, only NSP11-CoV aggregates do not show any positive signal near 200 nm, which corresponds to the β -sheet. This information is updated in the manuscript, and the respective figures and figure legends are accordingly modified.*

2. In the section “X-ray diffraction pattern (XRD) of aggregates” on page 13 of the revised manuscript, the authors need to comment on whether any XRD pattern was observed in the case of ORF10 protein aggregates.

Authors' Response: *We are thankful for this suggestion. We have not observed any XRD pattern for ORF10 protein aggregates which can be due to the formation of amorphous aggregates.*

3. Page 14: “The aggregates formed by NSP6-p exhibit a classic meridional XRD reflection at

~4.7 Å.” The authors need to mention why CD spectra were not recorded for the NSP6-p aggregates.

Authors' Response: NSPp-6 aggregates were washed with buffer twice to remove the DMSO content; however, the sample produced high noise while recording the CD spectra. Therefore, the spectra were not precise due to noise and are not included in the data. However, as suggested, we have mentioned this in the legend of Figure 9 of the revised manuscript.